# Hotspots of dendritic spine turnover facilitate clustered spine addition and learning and memory

Adam C. Frank[1,2], Shan Huang[1,2], Miou Zhou[1,2], Amos Gdalyahu[1,3], George Kastellakis[4], Tawnie K. Silva[1,2], Elaine Lu[1,2], Ximiao Wen[5], Panayiota Poirazi[4], Joshua T. Trachtenberg[1] & Alcino J. Silva[1,2]

Modeling studies suggest that clustered structural plasticity of dendritic spines is an efficient mechanism of information storage in cortical circuits. However, why new clustered spines occur in specific locations and how their formation relates to learning and memory (L&M) remain unclear. Using in vivo two-photon microscopy, we track spine dynamics in retrosplenial cortex before, during, and after two forms of episodic-like learning and find that spine turnover before learning predicts future L&M performance, as well as the localization and rates of spine clustering. Consistent with the idea that these measures are causally related, a genetic manipulation that enhances spine turnover also enhances both L&M and spine clustering. Biophysically inspired modeling suggests turnover increases clustering, network sparsity, and memory capacity. These results support a hotspot model where spine turnover is the driver for localization of clustered spine formation, which serves to modulate network function, thus influencing storage capacity and L&M.

[1] Department of Neurobiology; Integrative Center for Learning and Memory; Brain Research Institute, University of California, Los Angeles, CA 90095, USA. [2] Department of Psychology; Department of Psychiatry and Biobehavioral Sciences, University of California, Los Angeles, CA 90095, USA. [3] Department of Neurobiology, Tel Aviv University, Tel Aviv 69978, Israel. [4] Institute for Molecular Biology and Biotechnology (IMBB), Foundation for Research and Technology-Hellas (FORTH), GR, 70013 Heraklion Greece. [5] Department of Mechanical and Aerospace Engineering, University of California, Los Angeles, CA 90095, USA. Adam C. Frank and Shan Huang contributed equally to this work. Correspondence and requests for materials should be addressed to P.P. (email: poirazi@imbb.forth.gr) or to A.J.S. (email: silvaa@mednet.ucla.edu)

Recent findings indicated that memory storage processes operate conjointly at the level of neurons, dendrites, and dendritic spines[1–5]. Furthermore, dendritic spines are dynamic structures whose formation and elimination is postulated to expand memory storage capacity beyond that permissible solely from synaptic weight changes of existing synapses[4,6,7]. A variety of studies in varying preparations and organisms have shown that spine turnover is modified by electrical activity, sensory experience, and learning[8–13]. In addition, results from juvenile zebra finch show that endogenously higher levels of spine turnover before tutoring correlate with a greater capacity for subsequent song learning during the critical period[14]. Furthermore, spine structural dynamics[8,11,15–17] and activity[18] are thought to be clustered in dendrites. These findings support the hypothesis that clustering of plasticity events within dendrites is a means to efficiently store information[2,7,19,20]. However, although both spine turnover and spine clustering are shown to impact

learning and memory, it remains unclear how spine turnover and clustered spine addition relate to one and other and how these subcellular processes influence neural network function.

Here, we used transcranial two-photon microscopy to track spine dynamics, and examined the relation between basal spine turnover, contextual or spatial learning and memory, and subsequent spine clustering in the mouse retrosplenial cortex (RSC). The RSC is a neocortical structure conserved from rodents to humans that is critical for spatial and contextual learning and memory[21]. Lesions of RSC impair performance in the Morris water maze (MWM)[22–25] and contextual fear conditioning (CFC)[26]; inactivation of NMDA (N-methyl-D-aspartate) receptors in RSC impairs contextual fear memory recall[27] while blockade of AMPA (α-amino-3-hydroxy-5-methyl-4-isoxazolepropionic acid) receptors impairs Morris water maze performance; Overexpression of the transcription factor CREB (cAMP Responsive Element Binding Protein) in this structure

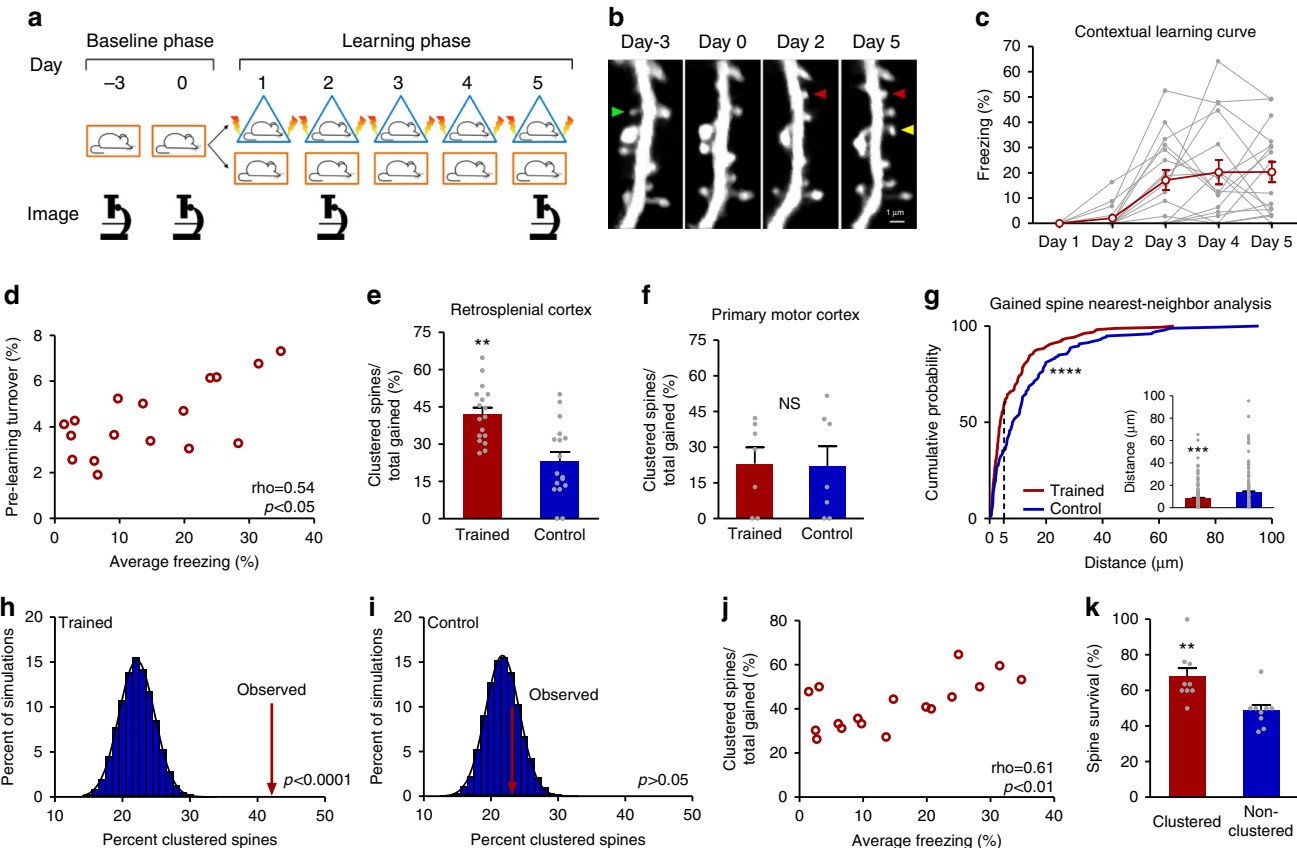

**Fig. 1** Pre-training dendritic spine turnover and learning-related spine clustering correlate with contextual learning. **a** Timeline of contextual learning and imaging. **b** Representative example of longitudinal imaging. Lost spine is denoted by green arrowhead. One gained spine is denoted by a red arrowhead; another gained spine (yellow arrowhead) is added by day 5 within 5 μm, thus forming a cluster. Scale bar indicates 1 μm. **c** Learning curve of mice during CFC. **d** Dendritic spine turnover before training correlates with future learning. Scatter plot shows spine turnover before training vs. average contextual freezing ($n = 17$ mice; Spearman's rho = 0.54, $p = 0.0255$). **e** Trained mice have a higher percentage of newly added spines that occur in clusters (≤5 μm) (Trained = 42.0% vs Control = 23.2%, $n = 17$ mice per group; Mann–Whitney $U = 51.00$, $p = 0.0014$). **f** Contextual learning does not increase spine clustering in primary motor cortex (Trained = 23.0% vs Control = 22.0%, $n = 7$ mice per group; $U = 24.00$, $p = 1.00$). **g** CFC significantly shifts inter-spine distances between gained spines toward smaller values. (Trained, $n = 155$ distance measurements; Control, $n = 173$ distance measurements; Two-sample Kolmogorov–Smirnov, $D = 0.2610$, $p = 2.1006e−05$). Inset is mean ± s.e.m. of values in distribution (8.3% vs 13.6%, Mann–Whitney $U = 10294$, $p = 0.0003$). **h** Clustered spine addition is significantly greater than chance. The histogram shows percent clustering from 10,000 simulations of randomized new spine positions; arrow represents spine clustering observed from data; black line is Gaussian fit (mean of Gaussian fit = 22.1%, observed = 42.0%, $n = 17$ mice; $p < 0.0001$). **i** In control mice, spine clustering is not greater than chance. Histogram is calculated as in **h** but using control mice data; arrow represents percentage of clustered spines observed from data; black line is Gaussian fit (mean of Gaussian fit = 21.7%, observed = 23.2%, $n = 17$ mice; $p > 0.05$). **j** Clustered spine formation is linearly correlated with freezing averaged from Day 2 to Day 5 ($n = 17$ mice; Spearman's rho = 0.61, $p = 0.0088$). **k** Clustered spines have a higher survival rate than non-clustered spines 4–6 weeks after training (67.6% vs 49.0%, $n = 9$ mice; Mann–Whitney $U = 8.000$, $p = 0.0042$). Data are represented as mean ± s.e.m. for **c**, **e**, **f**, inset of **g** and **k**. ****$p < 0.0001$, ***$p < 0.001$, **$p < 0.01$; NS not significant

results in enhanced spatial memory[28]. Immediate early gene expression studies further support a role for RSC in learning and memory by showing increases in c-fos and Arc gene expression following context exposure and contextual fear conditioning[29]. Finally, direct re-activation of the population of RSC cells active during contextual fear conditioning is sufficient to drive freezing responses and activate downstream neuronal circuits engaged during fear memory retrieval[30]. Thus, RSC is an ideal cortical structure in which to examine the effects of contextual and spatial learning on spine dynamics.

Here we show that pre-learning spine turnover predicts both learning and memory performance and learning and memory-related spine clustering. Accordingly, a genetic manipulation that enhances pre-learning spine turnover also enhances clustering and learning and memory. Furthermore, we find that pre-learning spine turnover and learning-related clustering are related processes that themselves exhibit spatial clustering within the dendritic tree. Finally, using biophysically inspired modeling, we find that turnover and clustering increase neuronal firing and network sparsity, thus enhancing memory capacity. We posit a hotspot model of spine formation in which higher rates of pre-learning spine turnover facilitate the formation of learning and memory-related clustered spines near regions of turnover, and that clustering serves as a means to stabilize structural plasticity.

## Results

**Spine turnover and clustering predict learning and memory.** We used in vivo two-photon microscopy to image spines in RSC in *Thy1-YFP-H* mice and coupled this with a CFC protocol (Fig. 1a, b) that produced a gradual increase in contextual freezing over 5 days of training (Fig. 1c). There was individual variability in the pre-learning spine turnover ratio (proportion of formed and lost spines) as measured in two imaging sessions before commencement of training. Strikingly, we found that this pre-learning turnover ratio correlated with levels of future contextual learning and memory (Fig. 1d) as well as learning rate (Supplementary Fig. 1a, c), though no association was found between spine turnover and animal age within the range tested (Supplementary Fig. 2; see Supplementary Fig. 3a, b for correlation of pre-learning turnover and freezing through training). Therefore, increased pre-learning spine turnover is associated with higher levels of learning and memory in mice—even when turnover is examined outside of developmental critical periods—indicating that spine turnover is an important determinant of learning and memory, not only during a critical period in zebra finch[14], but also in adult mammals.

To determine whether contextual learning affects spine dynamics, mice were split into two groups: one group underwent CFC while the second remained in their home cages. Both groups were imaged on the same schedule: in the early stages of learning and again at the end of day 5 of training (Fig. 1a). Comparisons between groups demonstrated that training had no impact on the rate of gain, loss, or turnover of spines (Supplementary Fig. 4a-c), such that both groups had similar numbers of spines at the start and end of the experiment (Supplementary Fig. 4d). However, trained animals showed a striking increase in the number of new spines that were clustered (two or more spines within 5 μm of each other; Fig. 1e). As the RSC is involved in processing context exposure alone, shock alone, and conjoint context exposure plus shock[29], we utilized primary motor cortex in a separate group of animals as an additional negative control. Specifically, primary motor cortex is not known to be involved in contextual fear conditioning[31] and thus context exposure and shock are not expected to influence spine dynamics in this brain region. In fact, we found that in primary motor cortex, animals trained in

contextual fear conditioning and home cage control animals had similar levels of clustering (Fig. 1f) suggesting that the changes we saw in RSC are specific to structures involved in contextual learning and memory. We found that clustered spine formation occurred throughout training, but was more frequent with additional days of training (Supplementary Fig. 5; additional properties of spine clustering, Supplementary Table 1 and 2). This finding suggests that as animals attain more information, there is a concomitant increase in spine clustering. We chose a 5 μm window for our analyses as a number of biochemical, electrophysiological, and structural studies suggest that a 5–10 μm distance between spines facilitates sharing of resources, spine co-activation, and learning-induced structural plasticity[16,18,32–36]. Furthermore, measurements of the nearest neighbor distances (the distance between a new spine to its closest new spine neighbor) for spines formed during learning were consistent with the results of the 5 μm analyses: in trained animals, the distribution of the nearest neighbor distances was significantly shifted towards smaller values (Fig. 1g). Furthermore, resampling analysis of the data from trained animals revealed that clustering within 5 μm would occur randomly for only 22.2% of newly added spines—consistent with the amount of clustering we observed in control animals and in motor cortex—while an average of 42.0% of new spines were clustered in RSC of trained animals (Fig. 1h and Supplementary Fig. 6a). Resampling analysis of data from untrained control animals revealed that clustering within 5 μm would occur randomly for 21.8% of newly added spines—consistent with random clustering calculated for trained animals—and near the observed value of 23.2% in control animals (Fig. 1i).

Mice that have higher rates of learning-related spine clustering in RSC exhibited more contextual freezing (Fig. 1j) and higher rates of learning and memory (Supplementary Fig. 1b, d; no association was found between conditioning, clustering, or baseline turnover in primary motor cortex, Supplementary Fig. 7; no association was found between clustering and freezing by day 2, while there is a correlation between clustering and freezing at day 5, Supplementary Fig. 3c, d; no association exists between spine gain and average freezing, Supplementary Fig. 8a; a positive correlation exists between spine loss during learning and average freezing, Supplementary Fig. 8b). Of note, there is a significant positive correlation between spine turnover rates before and during learning (Supplementary Fig. 8c), and as such a significant correlation between freezing and spine turnover during learning (Supplementary Fig. 8d). The linear relationship between spine clustering and contextual learning and memory highlights the potential role of clustered plasticity as a mechanism for cortical information storage. Consistent with the idea of sparse encoding, clustered spines formed during learning in RSC are on average only 2.5% of the total population of spines. Further supporting the hypothesis that clustered spine addition contributes to memory storage in CFC, we found that spines added in clusters in RSC during training have a higher survival rate compared to non-clustered spines (added during training) when examined 4–6 weeks after training (Fig. 1k); the survival rate of clustered spines is independent of the original cluster size (i.e., the number of constituent spines within a cluster, Supplementary Table 3). When analyzed at the level of individual clusters, the majority of clusters remain fully or partially intact, with an average of only 6.1% of clusters lost at 4–6 weeks (Supplementary Fig. 9). Altogether, these results indicate that clustered plasticity is a general information storage mechanism, not only for procedural memory in motor cortex[8], but also for episodic-like memory in RSC.

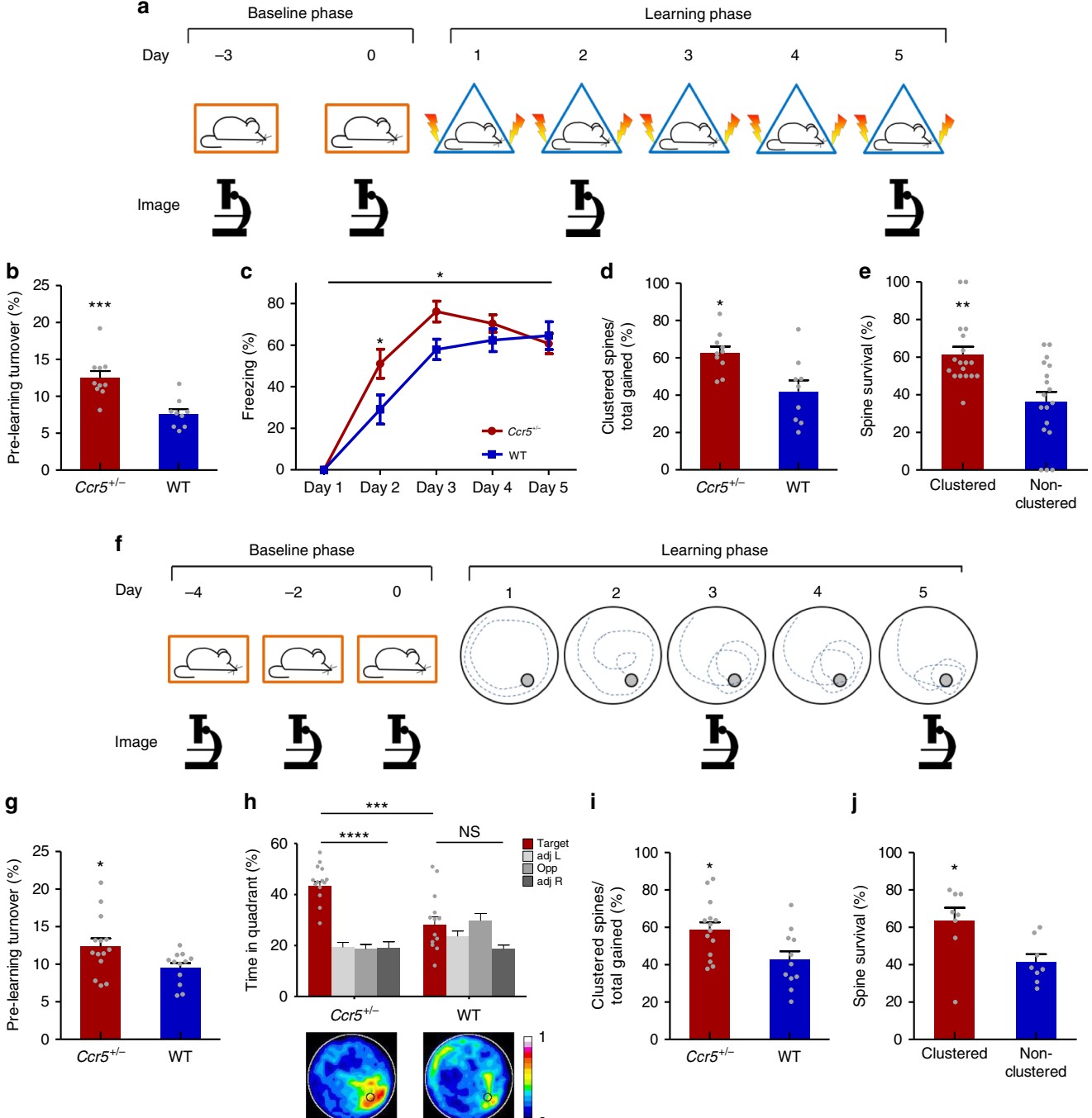

**Genetic enhancement of spine turnover increases memory.** Given the results described above, we were interested in genetically manipulating spine dynamics and studying the impact on learning and memory and clustering. To do so, we examined the effect of a null heterozygous mutation of the *C-C chemokine receptor type 5* ($Ccr5^{+/-}$) on basal spine turnover (Fig. 2a, f). Remarkably we found that pre-training spine turnover was enhanced in the RSC of $Ccr5^{+/-}$ mice (Fig. 2b, g). Chronic blockade of NMDA receptors with the antagonist MK801 prevented this increased spine turnover in $Ccr5^{+/-}$ mice, while having no effect on their wild type (WT) littermates (Supplementary Fig. 10a), demonstrating that the enhancement of pre-learning spine turnover observed in $Ccr5$ mutants is NMDA receptor dependent and that it may be due to plasticity-related

mechanisms. $Ccr5^{+/-}$ mice were recently shown to have enhancements in contextual and spatial memory[37]. Given this increased basal turnover, as well as enhanced contextual and spatial memory performance, we posited that learning-related spine clustering would also be enhanced. To test this hypothesis, we trained $Ccr5^{+/-}$ mice and their WT littermate controls in either CFC or MWM)—a spatial learning task—and imaged dendritic spines in RSC in a subset of animals that expressed Thy1-YFP (Fig. 2a, f). We confirmed that the $Ccr5^{+/-}$ mice showed superior performances in both CFC (Fig. 2c and Supplementary Fig. 11a) and MWM (Fig. 2h and Supplementary Fig. 11b, c). The results showed that training in MWM also induced clustered spine formation in RSC (Supplementary Fig. 12), and this clustering is significantly greater than chance

(Supplementary Fig. 13). Remarkably, contextual and spatial learning-related spine clustering were enhanced in the RSC of $Ccr5^{+/-}$ mice (Fig. 2d, i). Chronic treatment with MK801 impaired clustered spine formation in both $Ccr5^{+/-}$ and WT mice (Supplementary Fig. 10b), demonstrating that spine clustering is NMDA receptor dependent and that it may be a plasticity-dependent mechanism involved learning and memory. Consistent with the results presented above, clustered spines added during training (CFC or MWM) were significantly more stable than non-clustered spines at 4 weeks post-training for both $Ccr5^{+/-}$ and WT animals (Fig. 2e, j). Importantly, at 4 weeks post-training, the percentage of clustered spine survival correlated with remote memory performance (Supplementary Fig. 14), again suggesting a role for clustered spines in cortical information storage. Taken together, these results indicate that enhancements of spine turnover and spine clustering may be driving enhancements in contextual and spatial learning and memory.

**Spine clusters form within hotspots of spine turnover**. We have found that pre-learning spine turnover and learning-related spine clustering both correlate with contextual learning and memory. As expected, we also found a significant positive correlation between pre-learning spine turnover and learning-related spine clustering in both trained WT and $Ccr5^{+/-}$ mice (Fig. 3a, d). To further explore the spatial relation between these two phenomena within dendrites, we examined turnover and clustering at the level of dendritic segments and found that segments with greater amounts of pre-learning turnover also have increased levels of learning-related clustering (Fig. 3b, e and Supplementary Fig. 15). Next, we analyzed the distribution of dendritic segments regarding levels of pre-learning turnover and learning-related clustering. We found that the percentage of segments with con-cordant turnover and clustering (segments having both clustering and turnover or segments with neither) was higher than random chance, while the percentage of segments with discordant turnover and clustering (segments with only one of the two) was lower than chance level (Supplementary Fig. 16). Accordingly, we analyzed the average nearest-neighbor distance between learning-related clustered spines and spines having undergone pre-learning turnover and found this value to be significantly smaller than random chance (Fig. 3c, f and Supplementary Fig. 6b). Furthermore, the average distance between clustered spines and turnover spines is significantly smaller in trained animals relative to home cage controls (Supplementary Fig. 15c). Finally, we find that the average nearest neighbor distance from lost spines to new spines is significantly smaller for neighboring new spines that are part of a cluster (Fig. 3g). All together, these data demonstrate the presence of hotspots of turnover and clustered spine addition in

dendrites. We propose that pre-learning spine turnover may be used by neurons to sample their surrounding synaptic space. Increased turnover rate, may allow neurons to more frequently sample this space and thus optimize connectivity with appropriate presynaptic partners during learning. Clustering may then serve to stabilize these new synapses (Fig. 3h).

**Computational modeling of spine turnover impacts network**. We have demonstrated that pre-learning turnover predicts learning-related clustering, localization of clustering, and rates of learning and memory. Next, we explored how these processes might relate to neural network function. Using a biophysically inspired neuronal model[38] (parameters in Supplementary Table 4), and working under the framework of dendritic protein synthesis[32] and synaptic capture[39], we found that increased spine turnover predicted increased spine clustering following learning (Fig. 4a and Supplementary Fig. 17). In addition, we found that new spines which are added and consolidated after the first day of simulated training tended to be more clustered as spine turnover was increased (Fig. 4b), in line with our experimental findings. To assess the discrimination capacity of this network and its relation with spine turnover, we then serially encoded 10 memories in the same model network. As the number of dendrites with high synaptic turnover increased, the sparsity of the firing rates of memory engrams was also increased (Fig. 4c). This property could allow the network to discriminate between a larger number of memories and thus may increase overall memory capacity. Taken together, these results support the hypothesis that spine turnover is a driving mechanism for the localization of spine clustering, which in turn influences neuronal firing, memory discrimination, and therefore storage capacity.

## Discussion

Our understanding of how information is stored and memories formed within the brain has seen remarkable advancements recently[4,40]. For example, it is now generally accepted that information processing and storage occurs across physiological and morphological levels from neurons to dendrites to individual dendritic spines[1–5]. For instance, a recent study demonstrated that light-activated shrinkage of new or recently potentiated dendritic spines was sufficient to weaken memory strength[41]. However, we still lack a unifying framework for how subcellular processes, such as dendritic spine addition, affect cellular and network properties during learning and information storage.

An emerging theory, known as the clustered plasticity hypothesis, proposes that plastic events occur in a clustered fashion within the dendritic tree[2,7,19,20]. In agreement with the clustered plasticity hypothesis, several studies have shown that

**Fig. 2** $Ccr5$ heterozygous null mutation ($Ccr5^{+/-}$) augments pre-training and learning-related spine dynamics in RSC. **a** Timeline of CFC training and imaging. **b** $Ccr5^{+/-}$ mice have increased baseline spine turnover prior to CFC (Day −3 to Day 0) relative to WT littermates ($Ccr5^{+/-}$ $n = 10$ (12.5%), WT $n = 9$ (7.6%); $U = 6.00$, $p = 0.0006$). **c** $Ccr5^{+/-}$ mice show enhanced contextual learning and memory relative to WT littermates ($Ccr5^{+/-}$ $n = 12$, WT $n = 15$; Two-way RM ANOVA, genotype×time interaction: $F_{(4,100)} = 2.60$, $p = 0.0404$; Bonferroni post-test for Day2: $p < 0.05$). **d** The percentage of new spines added in clusters during CFC is significantly greater for $Ccr5^{+/-}$ vs WT ($Ccr5^{+/-}$ $n = 10$ (62.6%), WT $n = 9$ (42.1%); $U = 15.50$, $p = 0.0178$). **e** Clustered spines added during CFC are more stable at 4 weeks post-training than non-clustered spines (61.6% vs 36.4%, $n = 18$ mice with combined $Ccr5^{+/-}$ and WT; $U = 62.50$, $p = 0.0017$). **f** Timeline of MWM training and imaging. **g** $Ccr5^{+/-}$ mice have increased baseline spine turnover prior to MWM (Day −4 to Day −2) compared to WT ($Ccr5^{+/-}$ $n = 14$ (12.4%), WT $n = 12$ (9.5%); $U = 38.00$, $p = 0.0193$). **h** In a MWM probe test at 3 days of training, $Ccr5^{+/-}$ mice spent significantly more time in the target quadrant; at this point in training WT mice did not search selectively for the platform ($Ccr5^{+/-}$ $n = 14$, WT $n = 13$; Two-way RM ANOVA, genotype×percent time in each quadrant interaction $F_{(3,75)} = 9.11$, $p < 0.0001$; Bonferroni post-tests for target quadrant vs. all other quadrants: $p < 0.0001$ for $Ccr5^{+/-}$, $p > 0.05$ for WT; Unpaired $t$-test for target quadrant, $t_{(25)} = 4.173$, $p = 0.0003$). Heat maps below bar graphs show combined traces of mice from each group during the probe test. **i** The percentage of new spines added in clusters during MWM training is significantly greater for $Ccr5^{+/-}$ vs WT ($Ccr5^{+/-}$ $n = 14$ (58.7%), WT $n = 11$ (42.6%); $U = 33.50$, $p = 0.0186$). **j** Clustered spines added during MWM are significantly more stable at 4 weeks post-training than non-clustered spines (63.6% vs 41.5%, $n = 8$ mice with combined $Ccr5^{+/-}$ and WT; $U = 10.00$, $p = 0.0207$). Data are represented as mean ± s.e.m. Mann–Whitney $U$-test was used in **b**, **d**, **e**, **g**, **i** and **j**. ****$p < 0.0001$, ***$p < 0.001$, **$p < 0.01$, *$p < 0.05$; NS not significant

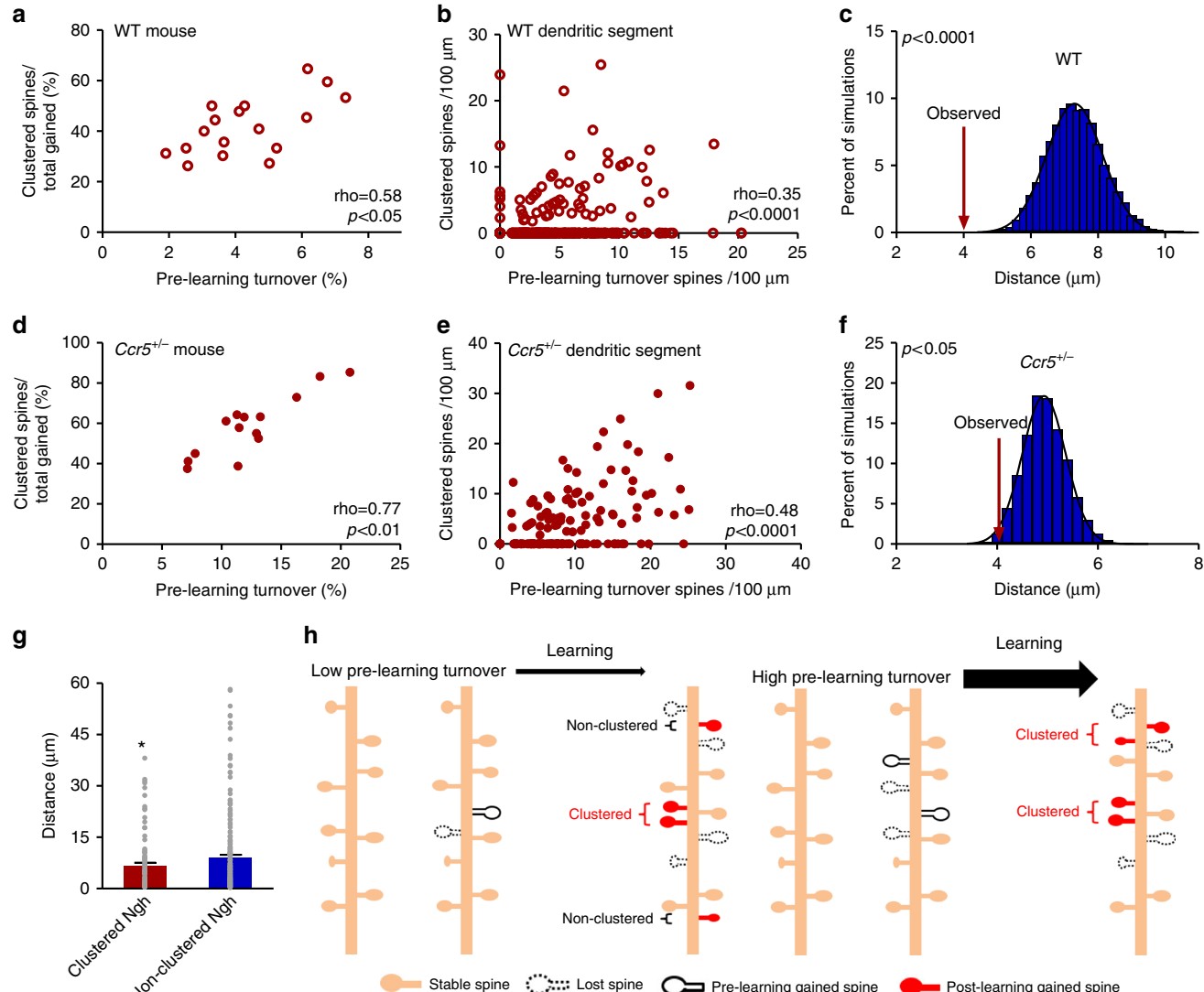

**Fig. 3** Learning-related spine clustering occurs within segments of increased pre-learning spine turnover. **a**, **d** A significant positive correlation exists between pre-learning spine turnover and learning-related spine clustering within individual WT (**a**, $n = 17$ mice; $p = 0.0154$) and $Ccr5^{+/-}$ mice (**d**, $n = 14$ mice; $p = 0.0014$). **b**, **e** A significant positive correlation exists between density of spines undergoing pre-learning turnover and density of learning-related clustered spines on each dendritic segment of WT (**b**, $n = 577$ segments across 17 mice; $p < 0.0001$) and $Ccr5^{+/-}$ mice (**e**, $n = 157$ segments across 14 mice; $p < 0.0001$). Clustering and turnover were normalized to dendritic length to facilitate comparisons across mice. **c**, **f** The average nearest neighbor distance (NND) from each learning-related clustered spine to the closest pre-learning turnover spine is significantly smaller than chance in both WT (**c**) and $Ccr5^{+/-}$ mice (**f**). 10,000 simulations of randomized pre-learning turnover spine positions were run, with NND from a clustered spine to a pre-learning turnover spine measured and averaged. Arrow represents the actual average NND (4.0 μm in WT, and 4.2 μm in $Ccr5^{+/-}$) observed in the data. Black line is Gaussian fit of data (WT: mean of Gaussian fit = 7.3 μm, $n = 152$ distance measurements over 17 mice; $p < 0.0001$; $Ccr5^{+/-}$: mean of Gaussian fit = 4.9 μm, $n = 322$ distance measurements over 14 mice; $p = 0.0293$, one sided). **g** Average distance between lost spines and gained spines is significantly smaller for clustered spine neighbors. NND were measured from each lost spine to the closest gained neighboring spine and this average distance is smaller if nearest gained spine is a part of a cluster (6.67 μm vs 9.09 μm, $n = 98$ clustered neighbor (ngh) measurements, $n = 214$ non-clustered neighbor measurements; Mann–Whitney $U = 8996$, $p = 0.0440$). **h** Model of clustered structural plasticity. Dendrites with higher rates of pre-learning spine turnover may allow neurons to more efficiently sample the surrounding synaptic space and subsequently establish more clustered connections. Clustered spine addition within a small spatial window allows for stabilization of the encoded information. Spearman's rho is indicated in **a**, **b**, **d** and **e**. *$p < 0.05$. Data are represented as mean ± s.e.m. in **g**. Mann–Whitney $U$-test was used in **g**

long-term potentiation (LTP) induces biochemical interactions between clustered spines that alter the threshold for induction of LTP[16,33,35]. Furthermore, facilitation of LTP between clustered spines is also due to the sharing of protein synthesis products[32]. Studies of structural plasticity have additionally shown that learning drives clustered addition of new dendritic spines[8,34]. From a functional perspective, recent data demonstrated enhanced orientation selectivity due to clustered synaptic inputs[18]. Interestingly, studies in sensory cortices also suggest that clustered spines may serve to integrate different inputs, as synapses on nearby spines appear to code for distinct visual orientations, sound frequencies, or whisker combinations[42–44]. However, the principles that govern where and to what extent clustered plasticity operates within dendrites, as well as how these subcellular events impact network dynamics have remained unclear.

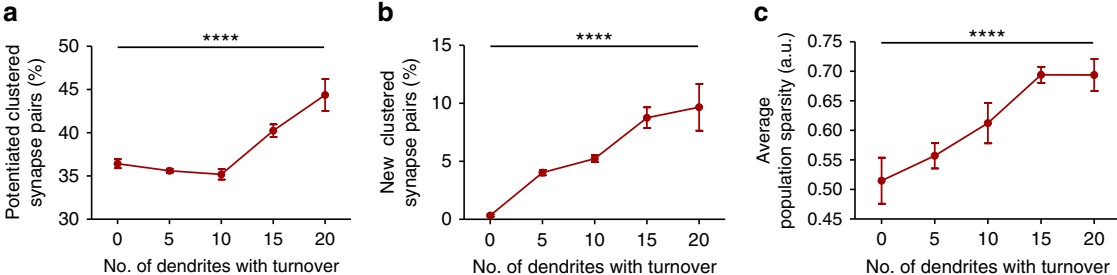

**Fig. 4** Spine turnover influences clustering and network functions in a computational model. **a** Increasing the number of dendritic branches that undergo high synaptic turnover increases the incidence of potentiated clustered synapse pairs (two synapses within 0.2 a.u.) ($n = 10$ simulation trials; $F_{(4,45)} = 16.69$, $p < 0.0001$; $m = 2.05$, $r^2 = 0.41$, $p < 0.0001$). Clusters were evaluated after 4 repeated learning episodes spaced 24 h apart. **b** Increasing the number of dendritic branches with turnover increases the percentage of new synapse pairs that occur in clusters ($n = 10$ simulation trials; $F_{(4,45)} = 14.19$, $p < 0.0001$; $m = 2.34$, $r^2 = 0.54$, $p < 0.0001$). Synapses were considered as new if they emerged after day 1 of simulated training. **c** Increasing the number of dendritic branches with turnover increases the average neural population sparsity of 10 memory engrams which are serially encoded in the same circuit, 24 h apart ($n = 10$ memory engrams; $F_{(4,45)} = 7.96$, $p < 0.0001$; $m = 0.05$, $r^2 = 0.39$, $p < 0.0001$). Sparsity was assessed by the Treves-Rolls sparsity metric. Data are represented as mean ± s.e.m. Second reported $p$-value is from the post test for linear trend following one-way ANOVA. ****$p < 0.0001$

Our in vivo imaging and computational modeling suggest answers to these outstanding questions. First, we have shown that animals have individual variability in their baseline rate of spine turnover, and that higher rates of this pre-learning spine turnover correlate with future levels of learning and memory. While similar findings of pre-learning spine turnover predicting future learning have been previously demonstrated in zebra finch during the critical period for song learning[14], our results are important in extending these findings to adult mammalian learning and memory. Critical period plasticity is generally regarded as unique during brain development[45]. However, our work shows that structural mechanisms of plasticity, important during this juvenile period of learning, also operate in adult mammals. Furthermore, our results show that spine dynamics are an important component of neuronal plasticity, and learning & memory, and that this spans both classes within the phylum Chordata as well as very different memory systems. Finally, existing studies demonstrate that spine addition and elimination operate on a multitude of time scales, ranging from hours to days depending on brain region and learning task[10,12,17,46–51], and our data from RSC add to this growing understanding of the rich diversity of spine dynamics within the mammalian brain.

We also show that learning of contextual and spatial tasks increases clustered addition of dendritic spines in RSC. Spine clustering in association with learning has been demonstrated in primary motor cortex during motor learning tasks[8] and in barn owl vestibular systems during prism adaptation[34]. Our data add to this body of evidence in support of the clustered plasticity model by showing that spine clustering also occurs in association with episodic-like learning, and is moreover positively correlated with learning and memory. Importantly, we showed that the $Ccr5^{+/-}$ mutation alters spine dynamics and causes enhancements in learning and memory. Moreover, our $Ccr5^{+/-}$ results show that a genetic manipulation that increases spine turnover also causes an increase in clustering. While other studies have shown that shrinkage of spines[41] negatively impacts memory, our positive manipulation of spine dynamics adds a critical line of convergent evidence in support to the clustered plasticity model.

The increases in pre-learning spine turnover and clustering by the $Ccr5^{+/-}$ mutation could be prevented by an NMDA receptor antagonist, suggesting that both spine turnover and clustering are due to plasticity-dependent mechanisms mediated by NMDA receptors. In contrast, NMDA receptor antagonists did not affect the increased rates of spine turnover in $Fmr1$ knockout mice, a mouse model of autistic spectrum disorder[52], demonstrating that they are not due to plasticity mechanisms mediated by NMDA

receptors. In addition, the $Ccr5^{+/-}$ mutation also causes increased clustered spine formation after learning. We showed that clustered spines are more stable and therefore this may facilitate information storage. As such, although $Ccr5$ knockout mice and $Fmr1$ knockout mice both show enhanced spine turnover[52–55], the two mutations result in opposite effects on L&M: while the $Ccr5$ mutation causes L&M enhancement[37], the $Fmr1$ mutation causes L&M deficits[56–60].

Given our finding that increases in turnover occur in association with increased spine clustering and learning, we explored a possible structural relation between these phenomena. We find that learning-related clustering preferentially occurs on segments of dendrites that have undergone pre-learning turnover. Strikingly, our simulation results show that clustered spine addition preferentially occur near areas of spine turnover on these dendritic segments. These data suggest that pre-learning turnover facilitates the process of clustering and thus generates hotspots of structural plasticity. We propose that increased baseline turnover allows neurons to efficiently sample their synaptic space, such that they can optimize synaptic connectivity during learning. When new connections do occur, clustering then integrates and stabilizes the acquired information.

Finally, using biophysically inspired models, we show that spine turnover impacts network function by increasing clustering and network sparsity during memory recall. The net effect of these network dynamics would be an increase in network storage capacity. These results, in conjunction with recent electrophysiological data[18,42–44], begin to uncover how subcellular processes such as spine turnover and clustering could intersect with physiology to influence neuronal firing and subsequent network function. Altogether, the results presented here support a hotspot model where spine turnover is the driver for localization and rate of clustered spine formation, which serves to modulate network function, thus influencing storage capacity and therefore L&M.

## Methods

**Subjects**. All experiments were approved by the guidelines established by the UCLA Animal Research Committee. Adult (3–8 months old) male and female Thy1-YFP-H mice were used for experiments in Figs. 1 and 3. Adult (3–8 months old) male and female $Ccr5^{+/-}$; $yfp^+$ double transgenic mice and their WT littermates $Ccr5^{+/+}$; $yfp^+$ mice were used for the spine imaging experiments in Fig. 2 ($Ccr5^{+/-}$ breeders were C57BL/6NTac; Thy1-YFP-H breeders were C57BL/6 J). $Ccr5^{+/-}$ and WT littermates, with or without Thy1-YFP were used for the behavioral enhancement experiments. Animals were kept on a 12:12-h light:dark cycle with food and water ad libitum. For MK801 experiments, $Ccr5^{+/-}$; $yfp^+$ double transgenic mice and their WT littermates $Ccr5^{+/+}$; $yfp^+$ mice were used for the spine imaging and trained with CFC task. Intraperitoneal injections of MK801 were

performed twice daily (0.25 mg/kg dissolved in saline)[52,61]. Injections started at 4 days before the first imaging day, and continued for 13 days until the last imaging was done.

**Surgery and cranial window implantation**. The procedure we utilized for window implantation has been described in detail[62]; briefly, mice were anesthetized with isoflurane, placed in a stereotaxic frame, and kept warm with a monitored heating pad. Custom cut coverslips (square, $2 \times 2$ mm) were cleaned in ethanol and sterilized. A square region of skull 2 mm in width was marked using stereotactic coordinates (RSC: center at bregma −2.5 mm AP). The skull was thinned with a dental drill and removed. After cleaning the surgical site with saline, the coverslip was placed on the dural surface and fastened with adhesive and dental acrylics to expose a square window of ~2 mm. Next, an aluminum bar with a threaded hole was attached to stabilize the mice during imaging sessions. Finally, mice were maintained on antibiotics during recovery and also given daily injections of carprofen and dexamethasone for 1 week to reduce inflammation. Mice are allowed to recover for 2 weeks before the first imaging session.

**Contextual fear conditioning**. Following recovery from surgery, mice were handled and habituated to transport cues for 2 weeks. On the first day after handling/ habituation mice underwent the first home cage baseline imaging session (day 3). 2 days later (day 0) mice underwent the second baseline imaging session. The following day half the mice were randomly selected to begin contextual fear conditioning using a multi-day paradigm. Animals were placed in conditioning chambers, 45 s later were given one 1.5-s 0.5 mA shock, and 10 s later were given a second shock of the same intensity and duration. Animals were removed from the conditioning chamber 2 min later and placed in their home cage. On day 2 mice were conditioned again as on day 1, except that 90 min after conditioning mice were imaged. Training continued as above for a total of 5 days. Each conditioning chamber (32 cm wide, 25 cm high, 25 cm deep) is equipped with stainless steel grid floor (36 rods, each rod 2-mm diameter, 8-mm center to center; Med-Associates, Inc., Georgia, VT) and stainless steel drop-pan. Chambers are scented with 100% isopropyl alcohol to provide a background odor. Each chamber is equipped with an overhead LED light source providing white light. Each chamber is connected to a solid-state scrambler, providing AC constant current shock, controlled via an interface connected to a Windows computer running Video Freeze (Med-Associates, Inc.), a program designed for the automated assessment of freezing, an index of fear memory. Learning rate for each mouse was calculated as the slope of the line connecting baseline freezing on day 1 before 1st shock and the asymptote level of freezing on the day this occurs. The asymptote level of freezing is the highest freezing that occurs chronologically before any decreases in freezing occur. Slope is then calculated as $m = $ (asymptote freezing−baseline freezing)/(day of asymptote freezing−day 0 of training). $Ccr5^{+/-}$ and WT littermates follow the same contextual fear conditioning and imaging protocol. Activity suppression ratio for each day is calculated as average activity during test divided by the sum of activity during baseline (activity on Day 1 before 1st shock) plus activity during test. Area under the curve is calculated in GraphPad Prism utilizing each animals fear learning curve.

**Morris water maze**. On the first day after handling/habituation (same as CFC) $Ccr5^{+/-}$ and WT littermates underwent the first home cage baseline imaging session (day 4). 48 h later (day 2) and 96 h later (day 0) mice underwent the second and third baseline imaging session. The following day, all mice with Thy1-YFP were trained with four trials per day for 5 days to find a hidden platform in Morris Water Maze. Each block consisted of two trials with 30 s interval between the trials. In each trial, mice were given 60 s to find the platform. If mouse found the platform earlier than 60 s in a certain trial, that trial terminated at the time mice finds the platform. If mice failed to find the platform, the trial terminated at 60 s. After each trial, mice were put on the platform for 15 s. Average latency (time spent in searching for platform) of four trials per day were analyzed and compared between days and between $Ccr5^{+/-}$ and WT littermates. On day 3 and day 5, probe tests with a time of 60 s were administered 1 h after training. During the probe test, platform was removed from the maze. (1) Percentage of time mouse spent in each quadrant and (2) platform crossing in each quadrant during probe test—which tests the accuracy of positional memory—were analyzed.

**Two-photon imaging**. A custom-built two-photon laser scanning microscope was paired with a Spectra-Physics 2-photon laser tuned to 920 nm. A $40 \times 1.0$ NA water immersion objective (Zeiss) was used to acquire images 90 min after each behavioral session. Mice were lightly anesthetized with isoflurane and attached to the head mount using a small screw. During the first imaging session, segments of apical dendrites from Layer V pyramidal cells were imaged. These segments were acquired within 200 μm from the cortical surface, likely representing dendrites located in layers I and II/III. Imaged segments were generally oriented in the $x, y$ plane of imaging with minimal z-projection. $512 \times 512$ pixel images were acquired at 0.5μm intervals to fully capture the segment of dendrite, and image stacks generally consisted of 20–30 slices. If a segment of dendrite was larger than could be acquired in one $512 \times 512$ stack, additional image stacks were sequentially

acquired through the $x, y, z$ plane of the dendrite in question so that its extent could be visualized. The same segments were repeatedly imaged across experimental days by locating their position via a coordinate system established during the first imaging session.

**Image and data analysis**. Dendritic spines were analyzed and counted by established criteria[62]. Specifically, the Spine Analysis software included in ScanImage was used to open all imaging days for a given segment of dendrite. A segment is classified as the entire visible length of a piece of dendrite; and segments were often followed across several images. The presence, gain, and loss of spines were quantified across days for each segment, and all segments were examined for a given animal. Distance measurements between spines occurred from the base of one spine to the base of the next spine and followed the shape of the dendritic shaft. Importantly, all images were coded following completion of the experiment so that the experimenter was blind to training status and genotype of all mice while analyzing and counting spines. A subset of the images was counted by two experimenters independently to confirm the results.

**Statistics**. Five and 3 separate replicates of imaging in RSC during CFC and MWM were run, respectively. The data from all replicates were pooled. Results for motor cortex CFC and imaging were collected from a single experiment. The size of each replicate was chosen to include approximately equal numbers of mice for group comparison, and to maximize the number of animals able to be imaged in one replicate cycle (~8–12 h). On the first day of training, every other cage was taken for behavior. Cage placement on the rack was random, choice of animals for CFC was as well. All available $Ccr5^{+/-}$ and their wildtype littermates at the age between 3 and 8 months were used for experiments. Correlations were calculated as Spearman's rho to compensate for the non-normality of the data. Similarly, the Mann–Whitney $U$-test was used for all other group comparisons, except where indicated. All $p$-values represent results from two-sided tests, except where indicated. Animals with behavioral data outside two standard deviations of the mean were excluded from statistical analysis. Animals with <5 spines formed during the learning phase were excluded from analysis. The turnover ratio equals the sum of the number of formed and lost spines between two time points divided by the sum of the total number of spines in each time point. The clustering ratio equals the number of clustered spines by total number of new spines formed after day 0 and stable at day 5. Clustered spines are defined as a new spine that has a distance <5 μm with another new spine. For analyses of the cumulative probability distributions of nearest neighbor measurements for newly formed spines and lost spines, the two-sample Kolmogorov–Smirnoff test was utilized.

For the resampling analysis of clustering, we have run simulations in two ways. First, in each resampling, the number of new spines added per segment of dendrite was used to pick an equivalent number of random positions along the same segment and assess whether these positions were within 5 μm of each other. When this was completed for all dendrites for a given animal the percent of clustered spines was calculated as the number of randomly selected new spine positions within 5 μm of each other divided by the total number of stably added new spines for that animal. In turn each animal's resampled clustering percentage was calculated and then these values were averaged together. This completed one resampling event, and this average value was plotted on the distributions shown in Fig. 1h for trained animals and Fig. 1i for controls. This process was then repeated for a total of 10,000 resampling events, which then gives the full distribution of random sampling. In our alternative approach, shown in Supplementary Fig. 6a, we utilized the number of new spines added to a segment of dendrite to select an equivalent number of positions along the dendrite at random, but only utilizing positions where a spine had been observed (whether previously observed or still present; i.e., the identity of a new spine was permuted). Here, the difference was that selected positions for each permutation were constrained to occur where spines were or had been in the dataset. This method was utilized to account for any unforeseen biological processes which might govern where spines can possibly exist (i.e., areas or regions of dendrite might exist that do not typically support new spine addition); for each resampling, by selecting positions at which spines occurred in our dataset we would then be assured that we did not inadvertently violate any processes that might govern potential spine locations.

For resampling analysis of average distance between clustered spines to spines undergoing turnover, both aforementioned methods were utilized. Specifically, the distribution shown in Fig. 3c, f was calculated by assessing the number of spines undergoing pre-learning turnover on each segment of dendrite and randomly selecting the same number of positions along this segment of dendrite; the distance from each clustered spine on the segment to the nearest randomly selected position was measured; these values were averaged for each animal and then all animals averaged together for each permutation, with the process repeated a total of 10,000 times. In the second method, shown in Supplementary Fig. 6b, positions along the dendrite were chosen at random but could only occur where a spine existed or had existed (i.e., permuted turnover spine identity), as described above for the resampling analysis of clustering.

For resampling analysis of distribution of segments with different levels of pre-learning turnover and post-learning clustering, the null hypothesis simulation is done by permuting the number of clustered spines on each dendritic segment

within each animal and recalculating the percentages of segments of the four categories. For example, a mouse has 10 segments and each segment has 2 numbers: number of clustered spines, and number of turnover spines. We simulate the null hypothesis by permuting the number of clustered spines on the 10 segments, without changing the original number of turnover spines on each segment. This will yield a random distribution of clustered spines on dendritic segments that is independent of pre-learning turnover. The permutation was repeated for 10,000 times.

**Computational modeling.** A previously published model network for memory allocation in neuronal populations was used to assess the role of synaptic turnover in memory[38]. The model consists of a network of excitatory and inhibitory neurons. Excitatory neurons are modeled as 2-layer units[63], consisting of a somatic spiking unit and 20 independent dendritic subunits capable of nonlinear synaptic integration, dendritic spike initiation and compartmentalized plasticity[7,64,65]. Each dendritic subunit integrates the incoming synaptic inputs which reside on it independently as follows:

$$\tau_b \frac{dV_b}{dt} = \sum_{i,j} w_j E_{syn}\, \delta(t - t_{i,j}) - V_b \qquad (1)$$

Where $V_b$ is the dendritic depolarization, $\tau_b$, $E_{syn}$ are constants (Model parameters and constants are listed in Supplementary Table 4), $w_j$ is the weight of synapse $j$ and $t_{i,j}$ are the timings of incoming spikes. Somatic spiking is given by an Integrate and fire model with adaptation[66]:

$$C \frac{dV}{dt} = -g_L(V - E_L) - g_{AHP}(V - E_K) + I_{syn}(t) \qquad (2)$$

$$\tau_{AHP} \frac{dg_{AHP}}{dt} = a_{AHP}\delta(t - t_{spike}) - g_{AHP} \qquad (3)$$

Where $V$ is the somatic voltage, $C$ is the somatic membrane capacitance, $g_L$, the leak conductance, $E_L$ the resting potential (0 mV), $g_{AHP}$ is the conductance of the adaptation (afterhyperpolarization, AHP) current and $E_K$ is the AHP reversal potential. $\tau_{AHP}$ is the adaptation time constant and $a_{AHP}$, the quantal increase of $g_{AHP}$ after a somatic spike which occurs at time $t_{spike}$. The synaptic current reaching the soma $I_{syn}$ is given by

$$I_{syn}(t) = g_{syn} \sum_n (V_{b,n}(t)) - IPSC(t) \qquad (4)$$

where $IPSC(t)$ is the total inhibitory input that the neuron receives and $g_{syn}$ is the dendritic coupling constant. Somatic spiking occurs when the somatic voltage reaches the spike threshold $\theta_{soma}$. The backpropagating action potential (bAP) is modeled by a depolarization component $V_{bAP}$ which is added to the depolarization of all the dendritic subunits. $V_{bAP}(t)$:

$$V_{bAP}(t) = E_{bAP} e^{-\frac{t}{\tau_{bAP}}} \qquad (5)$$

$E_{bAP}$ is the peak of the backpropagating depolarization and $\tau_{bAP}$ is the time constant of the bAP.

Calcium influx $\Delta C_{syn}$ near a synapse after a presynaptic spike is dependent on the depolarization of the dendritic subunit using a sigmoid rule that mimics the voltage dependence of the NMDA receptors as follows:

$$\Delta C_{syn} = a_{Ca} \frac{1}{1 + \exp\left(-\frac{V - 30\,mV}{5\,mV}\right)} \qquad (6)$$

where $\alpha_{Ca}$ is the maximum Ca$^{2+}$ influx and $V = V_b + V_{bAP}$.

Synapses are initially allocated randomly in dendritic subunits, given a random position between 0.0 and 1.0 in arbitrary units, and initial weight $w_{init}$. Calcium influx in a synapse during stimulation is the determinant of plasticity following the synaptic tagging and capture model[39]: Low-to-intermediate levels of calcium after stimulus presentation lead to the generation of a depotentiation synaptic tag while high levels of calcium lead to a potentiation tag (see Supplementary Table 4). The consolidation of synaptic tags into the weight of synapses is dependent on the level of plasticity-related-proteins (PRPs). The level of PRPs near a synapse is increased to its maximum value (1.0) when the calcium level at the synapse exceeds the threshold $\Theta_{PRP}$ and decays exponentially with time constant $\tau_{PRP}$. During the consolidation phase of LTP, tagged synapses capture proteins from all neighboring synapses which are at a distance of up to 0.2 a.u. away. This sum of available proteins determines the rate of consolidation of synaptic tags into the permanent weights $w$ of synapses. Considering that dendritic subunits correspond roughly to the size of oblique dendrites capable of independent dendritic integration (~ 50 μm), the distance of 0.2 a.u. translates to ~ 10 μm, a distance which is known to facilitate synaptic cross-talk[33]. Synaptic weights are subject to homeostatic plasticity, which normalizes the total synaptic input to a neuron over long time scales:

$$\frac{dw_j}{dt} = \frac{1}{\tau_H}\left(1 - \frac{\sum_j w_j}{w_{init} N_{syn}}\right) \qquad (7)$$

where $w_{init}$ is the initial synapse weight of synapses (0.2), $N_{syn}$ the total number of synapses in the neuron and $\tau_H$ the time constant of homeostatic synaptic scaling.

Synaptic turnover was simulated via the removal of synapses which have not been potentiated beyond their initial weight (0.2) in each neuron and the addition of an equal number of synapses from random presynaptic inputs which contact random postsynaptic dendritic subunits in the same neuron. The rate of synaptic turnover for each dendritic subunit can have two levels, a low turnover rate (0.1) or a high turnover rate (1.0). In the simulations shown in Fig. 4, the number of dendritic subunits with high turnover rate was increased from 0 to 20. Synapses were removed stochastically with probability (turnover rate)×$\Theta_{removal}$.

In the first set of simulations a memory was encoded in the neuronal population via encoding episodes which occur in 4 consecutive days and consist of activation of the memory-encoding presynaptic inputs for 4 s. After encoding, the positions of potentiated synapses only (synapses with weight > 0.8) on their respective subunits were used to identify the pairs of synapses which were clustered (were less than 10 μm from each other). The ratio (number of clustered potentiated synapse pairs)/ (total number of synapse pairs) was used to assess the degree of clustering shown in Fig. 4a. Synapses which were added after the first simulated day of encoding were considered as "new" synapses and the ratio (number of pairs of clustered new synapses)/(total number of synapse pairs) was used to assess the clustering level of new synapses (Fig. 4b). The complete distributions of the distances of potentiated synapse pairs for increasing numbers of high turnover dendritic subunits is shown in Supplementary Fig. 17.

In a second set of simulations, 10 different memories were encoded in the same population with 1 day inter-stimulus interval. Each memory was represented by a different set of presynaptic inputs. After encoding, the memories were re-activated and the firing rates of the excitatory population were used to assess the sparsity of the response to each memory using the Treves-Rolls sparsity metric[67] shown in Fig. 4c. For additional details of the model, please see reference[38]. The parameters of the computational model are listed in Supplementary Table 4. Simulation code was written in C++ and is available through the ModelDB database.

**Data availability**. The spine images and data that support the findings of this study are available from the corresponding author upon request. Spine resampling and randomization simulation codes were written in R 3.2.0 and can be accessed from corresponding author upon request. The computational model is available in ModelDB with accession number 227087.

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

## Acknowledgements

We wish to thank D. Buonomano, C.Portera-Cailliau, D. Cai, M. Handcock, M. Sehgal, N. Wisniewski and J. Shobe for critical comments on earlier versions of this manuscript. This work was supported by grants 2RF1AG013622-21, and the Dr. Miriam and Sheldon G. Adelson Medical Research Foundation to A.J.S., grant T32 MH19384-14 for A.C.F., and China Scholarship Council to S.H. and the ERC Starting Grant dEMORY ERC-2012-StG-311435 to P.P.

## Author contributions

A.C.F., S.H., A.G., J.T.T., and A.J.S. designed the longitudinal imaging experiments. A.C.F., S.H., and A.J.S. designed the behavioral paradigm. A.C.F. performed WT contextual conditioning experiments and associated imaging and data analyses. S.H. and A.C.F. performed the $Ccr5^{+/-}$ and WT littermates spatial and contextual experiments, associated imaging and data analyses. S.H. and A.C.F. performed the analysis of dendritic segments. A.G. helped with spine analysis. A.C.F., S.H., and X.W. performed the permutation tests. S.H. and E.L. performed MK801 experiments and associated imaging and data analyses. P.P. designed and G.K. performed modeling experiments. M.Z. and T.K.S.

first characterized the learning and memory enhancement of the $Ccr5^{+/-}$ mice. A.J.S. supervised the project. A.C.F., S.H., J.T.T., and A.J.S. wrote the manuscript.

## Additional information

**Competing interests:** The authors declare no competing financial interests.

