## [Peer Review File · Nature Communications]

Reviewers' comments:

Reviewer #1 (Remarks to the Author):

Many recent works suggest that clustered plasticity of dendritic spines is a potential mechanism for information storage in cortical circuits. In this study, Frank et al. followed the spine dynamics in the retrosplenial cortex (RSC) before, during and after two types of learning (contextual fear conditioning or CFC, and Morris Water Maze or MWM) to better understand the source and pattern of this phenomenon. They found that intrinsic spine turnover before learning is predictive of the learning performance, and the location/rates of spine clustering. They also show that mutant mice with higher intrinsic spine turnover have better learning and more clustered spine formation. Finally, the authors used computational modeling to explain the phenomena and proposed a 'hotspot' model in which spine turnover drives clustered spine formation.

This paper investigated a very important question in neuroscience. The proposal of synaptic clustering for information coding is really exciting. The experimental design is straightforward, and the paper is succinct. Nevertheless, much additional information is necessary to clarify and strengthen it. In particular, the authors need to clarify the assumptions and procedures of the computational analysis, and present more quantitative characterizations of the phenomena.

1. The authors need to better introduce and establish the role of RSC in the two learning paradigms. Do the two learning paradigms involve similar brain circuits? Does the RSC receive inputs from the same sources in the two learning paradigms?
2. Suppl. Table 1 & 2 seem to summarize only the data from CFC experiments. What about the data from the MWM experiments? Have the authors run the same permutation test on the spine data from MWM experiments? This is an important issue, because it will determine whether the learning rule found in this work can be generalized to different learning paradigms.
3. It is problematic to use mice kept in home cage as the only control for CFC. It is necessary to establish that the observed effects are not due to exposure to the context, or due to foot shock per se.
4. Do the learning-associated spine clusters disassemble during fear extinction? This will corroborate the notion that spine clustering is correlated with learning.
5. Earlier works showed that fear conditioning increases spine elimination in the frontal cortex (Lai et al., Nature 2012), and auditory fear conditioning increases spine formation in the auditory cortex (Yang et al., Nat. Neurosci. 2016). These results contrast with the report in this manuscript that spine dynamics does not change during CFC. How do the authors explain this discrepancy? Is the frontal cortex involved in CFC? If so, do the authors anticipate increased spine elimination as reported by Lai et al., or do they expect similar result as in RSC? This is an important issue, because it will determine whether the learning

rule found in this work is peculiar to RSC or generalizable to other brain regions involved in CFC.

6. When clustered new spines get eliminated after 4 – 6 weeks, do the whole cluster of spines disappear together? It will be informative if the authors can classify the pattern of elimination of clustered spines.

7. The authors claim that pre-learning spine turnover correlates with learning rate. Given the approximately sigmoidal shape of the learning curve shown in Suppl. Fig. 1a, it seems that the definition of the learning rate (slope) does not capture this feature, and it may be sensitive to fluctuations in freezing percentage after the animal's performance has almost plateaued. The authors may try to fit each mouse's learning curve with a sigmoidal curve and use the fitting parameters to correlate with learning performance.

8. The permutation test for spine clustering is problematic.

1) The first permutation test is invalid. The authors state that the "new spine" identity was randomly assigned to "the positions of all spines identified during imaging." What is the rationale for this procedure? Why should the "new spine" identity be assigned to a stable spine or even an eliminated spine?

2) It is unclear from the Methods section how the authors obtain the null distribution as shown in Fig. 1f, Fig. 3c, and Suppl. Fig. 6. According to the authors, the permutation test was done per dendritic segment. As spine density/distribution and the percentage of spine formation vary from segment to segment, the probability distribution of clustered spine formation will also differ from segment to segment, because with higher density of new spines there will likely be more clusters even under the null hypothesis. Thus it is invalid to pool the permutation results from all segments into a single histogram, and compare that with the overall percentage of clustered new spines.

3) The second method of permutation test (Suppl. Fig. 6) seems reasonable. However, it yields a distribution of percentage of clustered new spines that appears inconsistent with the spine density and spine formation rate given in Suppl. Table 1 and Suppl. Fig. 3a. Take D0-2 control mice for example. The average spine density is ~ 0.4 spine/ μm , which means the average separation between spines is ~ 2.5 μm . The new spines gained from D0 to D2 account for $\sim 4\%$ of total spines on D0, which means that, if the new spines are distributed uniformly and independently along the dendritic segment as the authors assumed in the null hypothesis, their average distance would be ~ 2.5 $\mu\text{m}/4\% = 62.5$ μm . Under the null hypothesis, the new spines along the dendritic segment follow a Poisson distribution, and the inter-spine interval follows an exponential distribution with parameter $\lambda = 1/(62.5$ $\mu\text{m})$. Therefore, the probability that the nearest neighbor of a spine is within 5 μm (i.e., they form a cluster) is $1 - \exp(-5/62.5) \sim 7.7\%$, which is approximately the same as the number for trained mice given in Suppl. Fig. 4 and much higher than the number for control mice given there.

Based on the same argument, if we take the total spine formation from D0 to D5 to be $\sim 8\%$ (Supple. Fig. 3) and ignore the complication caused by the fact that some spines formed

during D0-2 were lost by D5, we will have the probability of clustering (again, irrespective of whether the spine was formed during D0-2 or D2-5) as $\sim 1 - \exp(-5/31.25) \sim 14.8\%$. This number, although an over-estimation, is much lower than that for control mice (29%, Suppl. Table 1).

The authors need to give more details about the parameters and procedures used in the permutation test to clarify such discrepancies. If the control mice already exhibit significantly more clustering than expected from the null hypothesis, it would have a major impact on how the data for trained mice should be interpreted.

9. The idea that hotspots of spine turnover have higher clustered spine formation is very interesting. However, there are a number of issues with the analysis.

1) In this analysis, do spines that are eliminated count towards the "spines having undergone turnover"?

2) In Fig. 3b there are apparently 3 categories: segments with pre-learning turnover but without learning-related clustered spines (data points on x-axis), segments with learning-related clustered spines but without pre-learning turnover (data points on y-axis), and segments with both. How many segments are there in each category? Does the distribution among these categories differ from the null hypothesis, i.e., clustering is independent of pre-learning turnover?

3) How did the authors generate the null model for Fig. 3c? What does "random chance" mean in this simulation?

4) In control mice there are also clustered spine formation. Do these clusters also tend to form near sites with previous turnover? What is the average distance to nearest turnover site in control mice? Is it comparable or significantly different from the distance in learning mice?

5) Are sites of clustered spines themselves more dynamic? If so, this would imply that hotspots tend to persist.

10. The authors need to provide much more details about the biophysical model (Fig. 4) for the reader to understand how it works and how it is related to other results presented in this manuscript.

Minor issues:

1. The term "gained spines" is confusing, as it could mean spine formation, or the net number of spines gained (i.e., formation minus elimination).

2. The actual values of spine dynamics are often shown as plots but not explicitly given in numbers.

3. Is there any mouse that was trained but failed to learning the task? If so, did they exhibit

similar spine dynamics/clustering as control mice?

Reviewer #2 (Remarks to the Author):

Frank and colleagues follow dendritic spine dynamics in retrosplenial cortex during contextual fear conditioning and Morris Water Maze learning. They find that both the amount of pre-learning spine turnover as well as clustering of new spines during learning are correlated with learning performance. They further find that a transgenic mouse model that has previously been demonstrated to show enhanced learning also has higher pre-learning spine turnover and more learning-related clustered new spines.

This is an interesting and timely study, the data seem solid and mostly carefully analyzed. My strong concern is the model, which is very opaque. It is difficult to judge its quality and validity, as far too little information is provided: how exactly was the model adapted for the study? What had to be changed to fit the data? The presentation of the model results seems misleading. The methods say: "Clustering was defined as the percentage of dendritic branches containing 3 or more potentiated synapses." This would mean that there is simply more synaptic input, not more clustered input, which is trivial as spine turnover was increased? The authors should plot percentage of clustered new synapses of total gained synapses as in Figure 1d and g, or if the model does not contain addition of new spines, percentage of clustered potentiated synapses of all potentiated synapses within branches. But if the model only contains potentiation of synapses rather than addition of new synapses, and/or if there is no modelling of the spatial arrangement of synapses within branches, how does the model relate to the presented data? In my opinion it would be better to remove the model in its present state, the experimental data are strong enough on their own.

Further points:

- The analysis in Figure 3b is not suited. Because turnover and clustering are related at the level of individual mice, and mice have different levels of turnover/clustering, plotting the values of individual dendritic segments of all mice together without z-scoring will automatically result in a positive correlation. If the authors want to make the point that that individual dendrites with more turnover also have more clustering, it is imperative to show this WITHIN mice, for instance by z-scoring to normalize data between mice.

- Do animals with more pre-learning spine turnover also have increased spine gain/loss during learning? Is learning performance therefore related to spine gain, loss or turnover during learning? This is important to be able to make the point that it is really the PRE-LEARNING turnover that is important for learning.

- Is the turnover and clustering only related to the speed of learning or also to how well mice do at the end of training? Please plot the data in Figure 1c and g separately for freezing at day 2 and day 5.

- Related to the above question: please provide a time course of performance over learning for the Morris water maze for WT and transgenic mice.

- Clustering analysis: how was the distance between spines determined? Is it the distance between spine bases along the dendrite? Please add to methods.

Reviewer #3 (Remarks to the Author):

In the present manuscript, Frank, et al. reported learning-related clustering of spine in the retrosplenial (RSC) and its relationship with pre-learning turnover of spines using thy1-YFP expressing H-line mice. They found that the fear conditioning was more effective in those animal with higher pre-learning spine turnover, and induced clustered spine generation, which occurred more frequently in the better learning mice. The authors test the relationship using Ccr5 hetero KO mice (Ccr5) which showed rapid fear conditioning. They found both pre-learning spine turnover and post-learning-clustering were enhanced in Ccr5. Clustered new spines were more persistent than non-clustered one in WT and Ccr5. The authors confirmed the result with the Morris water maze. The clustered spine generation was more frequently detected in those mice with larger pre-learning spine turnover in WT mice. These results could be explained by their model in a previous study (Kastellakis et al., Cell Rep. 2016). The model also predicted, more sparse coding and less overlap of neurons in a neuronal network. The authors conclude such hotspot serves to modulate network function, and influencing storage capacity and learning and memory. These observations are intriguing and valuable to the field, provided that the major problem below is experimentally resolved.

The authors overlooked the fact that spine turnovers were increased in many animal models of ASD, where learning is impaired (ex. Pan et al. PNAS 107:17768, 2010; Padmashri et al. J. Neurosci. 33:19715, 2013, Isshiki et al. Nat Commun 5:4742, 2014; Jiang et al. J. Neurosci. 33:19518, 2013). These are opposite to the finding in this manuscripts. Thus, the authors need to reconcile their observations with the existing knowledge. One key observation is that spine turnovers are caused not only by plasticity, which is activity-dependent, but also by intrinsic fluctuations of spine structures, which are independent of any neuronal firing and plasticity mechanisms (Yasumatsu et al, J. Neurosci. 28:13592, 2008; Minerbi et al. PLoS Biol. 7:e1000136, 2009; Loewenstein et al. J. Neurosci. 35:12535, 2015; Dvorkin et al. PLoS Biol. 14:e1002572, 2016). Increased spine turnovers seen in ASD models were ascribed not to plasticity but to fluctuations in the case with FMRP1-KO mice (Nagaoka et a., Sci. Rep. 6:26651, 2016). This work showed that plasticity-induced spine turnovers were effectively prevented by intraperitoneal infusion of MK801 twice daily (0.25 mg/kg).

To resolve the major discrepancy, the authors should examine whether the MK801 treatment restores the enhanced spine turnovers in Ccr5 mice during the pre-training period, and those in WT during learning tasks as a positive control. If the increase in the

turnovers are restored by MK801, it shows increase in baseline learning, and is consistent with the authors interpretations. Such results also provide a way to understand why spine turnovers induce both the impairment and enhancement of learning in ASD and Ccr5 mice, respectively. The authors should provide a full introduction and explanation to the apparent discrepancy between ASD model and Ccr5 mice. If the increased turnovers might not be blocked by the MK801 treatment, the authors need to find other ways to consistently interpret their data sets and existing references.

Minor points:

- 1) The size of clustering is only qualitatively described in line 76 (two or more spines within 5um of each other) and rest of the manuscript. Could the authors be more quantitative about the size of cluster, for comparison in the future experiments, as the cluster is the major concern in this manuscript?
- 2) In figure 3, the authors examined only WT data, but the same analysis should be performed for Ccr5 mice.
- 3) In figure 4, the authors try to explain the experimental data of real dendritic spines with a theoretical model where only very simple mechanisms are incorporated into small circuits. Just for example, the model ignores the existence of synapse fluctuations. In other words, only way to account for the increase in spine turnover in this model is plasticity. Another point is that Fig. b-d must be pure predictions. In order to distinguish experimental data from the simplistic model predictions, the authors may consider moving Fig. 4 to supplementary figure, particularly because the model does not incorporate the fluctuations, although I think the model simulation is still valuable.
- 4) In line 32 and 160, the authors cite a paper (*Annu Rev Physiol* 55:397, 1993) for spine plasticity, but it gives a kind of historical consideration on structural plasticity, mainly of presynaptic terminals, and may be out of place. There is a more appropriate review, where notions of plasticity and intrinsic fluctuations of dendritic spines are consistently described (*Trends Neurosci.* 33:121, 2010).

Reviewers' comments:

Reviewer #1 (Remarks to the Author):

Many recent works suggest that clustered plasticity of dendritic spines is a potential mechanism for information storage in cortical circuits. In this study, Frank et al. followed the spine dynamics in the retrosplenial cortex (RSC) before, during and after two types of learning (contextual fear conditioning or CFC, and Morris Water Maze or MWM) to better understand the source and pattern of this phenomenon. They found that intrinsic spine turnover before learning is predictive of the learning performance, and the location/rates of spine clustering. They also show that mutant mice with higher intrinsic spine turnover have better learning and more clustered spine formation. Finally, the authors used computational modeling to explain the phenomena and proposed a 'hotspot' model in which spine turnover drives clustered spine formation.

This paper investigated a very important question in neuroscience. The proposal of synaptic clustering for information coding is really exciting. The experimental design is straightforward, and the paper is succinct. Nevertheless, much additional information is necessary to clarify and strengthen it. In particular, the authors need to clarify the assumptions and procedures of the computational analysis, and present more quantitative characterizations of the phenomena.

1. The authors need to better introduce and establish the role of RSC in the two learning paradigms. Do the two learning paradigms involve similar brain circuits? Does the RSC receive inputs from the same sources in the two learning paradigms?

We thank the reviewer for the opportunity to better review the role of the RSC in CFC and MWM. We have added an additional paragraph in our introduction that highlights a number of studies that implicate RSC in CFC and MWM. These studies used a variety of techniques, from immediate early gene expression to transient glutamatergic blockade to optogenetics.

2. Suppl. Table 1 & 2 seem to summarize only the data from CFC experiments. What about the data from the MWM experiments? Have the authors run the same permutation test on the spine data from MWM experiments? This is an important issue, because it will determine whether the learning rule found in this work can be generalized to different learning paradigms.

We have added an additional column of data to Supplementary Table 1 and 2 that contains information regarding spine properties and clustering properties for animals that underwent Morris Water Maze training. Further, we performed resampling analysis on our MWM data (see revision, Supplementary Fig. 13), which is consistent with results from CFC.

3. It is problematic to use mice kept in home cage as the only control for CFC. It is necessary to establish that the observed effects are not due to exposure to the context, or due to foot shock per se.

We appreciate the importance of determining whether the effects of training on spine dynamics observed in RSC are the result of the behavioral manipulations used and not a spurious procedural artifact. The RSC is an interesting and exciting cortical region that is intimately involved in processing spatial information. Additionally, Robinson *et al* used immediate early gene expression studies to show that the RSC is activated not only by CFC, but also by context exposure and even foot shock alone (Robinson et al., 2012). Given that RSC is engaged by context exposure and foot-shock alone, we could not use these manipulations as controls. Thus, to demonstrate that clustering observed in RSC following CFC was specific, we imaged primary motor cortex, a region known not to be involved in CFC, and found no difference in clustering between trained and home cage controls. We have highlighted this additional negative control by presenting this result as Figure 1f.

4. Do the learning-associated spine clusters disassemble during fear extinction? This will corroborate the notion that spine clustering is correlated with learning.

We thank the reviewer for this suggestion. However, there is a long literature that indicates that fear extinction involves new learning (not necessarily the loss of previous learning; Izquierdo et al., *Physiol Rev*, 2016), extinction is not a weakening of synapses potentiated during learning (Nabavi et al., *Nature*, 2014; Herry et al., *Eur J Neurosci*, 2010), and new spines are usually gained after extinction (Lai et al., *Nature*, 2012). Therefore, although the experiment suggested by the reviewer is interesting and worth doing, it could not be used as a control manipulation to test the function of clustering, and consequently, it is outside of the scope of our manuscript.

5. Earlier works showed that fear conditioning increases spine elimination in the frontal cortex (Lai et al., *Nature* 2012), and auditory fear conditioning increases spine formation in the auditory cortex (Yang et al., *Nat. Neurosci.* 2016). These results contrast with the report in this manuscript that spine dynamics does not change during CFC. How do the authors explain this discrepancy? Is the frontal cortex involved in CFC? If so, do the authors anticipate increased spine elimination as reported by Lai et al., or do they expect similar result as in RSC? This is an important issue, because it will determine whether the learning rule found in this work is peculiar to RSC or generalizable to other brain regions involved in CFC.

We thank the reviewer for highlighting the exciting and highly relevant work that examines learning related spine dynamics in frontal and auditory cortex. Consistent with spine imaging literature regarding experience and learning-related spine dynamics (Holtmaat 2006, Xu 2009, Wilbrecht 2010), Yang et al. see an increase in spine formation in auditory cortex associated with auditory fear conditioning. However, it is interesting to note that this selectively occurs at three days after auditory fear conditioning, but not at 2 hours post-conditioning. Lai et al. find an interesting connection between extinction of auditory fear conditioning and spine elimination within frontal cortex, though here the loss of spines associated with fear extinction is observed at 2 days after training. Other imaging literature shows spine gain in motor cortex 1 hour after attainment of the reaching task performance goal (Xu et al. 2009); Trachtenberg et al. find increases in the number of gained spines in barrel cortex 1 day after checkerboard whisker trimming (Trachtenberg 2002); Graham et al. find increases in spine gain in barrel cortex at 1 day after whisker stimulation (Graham 2002); Sanders et al. find a decrease in spine density 1 day after fear conditioning in CA1 neurons specifically activated during training (Sanders 2012). These results highlight the heterogeneity in both timing of imaging and in which cortical region spine dynamics have been examined in relation to learning and sensory stimulation. RSC is a uniquely positioned cortical area with connections to sensory and hippocampal-related areas and has only recently been studied using chronic in vivo imaging, though not directly in relation to learning (Raybuck 2017). It is possible that RSC does have transient increases in gained spines that are missed when animals are imaged 2 days after the first learning episode, as we have done here.

Concerning the circuits and brain regions involved in fear conditioning, Izquierdo et al. recently published an extensive review of the topic (Izquierdo et al 2016). Ventromedial prefrontal cortex (vmPFC) is involved in contextual fear conditioning and it is an interesting and open question as to how spines in this location would be influenced by CFC. Our studies would suggest that spine clustering might also occur in this region while animals undergo CFC. With regard to the findings of Lai et al., it is possible that the neurons examined are involved in learning the extinction rules and thus the gain of spines during extinction training is consistent with the aforementioned studies showing spine formation during learning; though we concur with the authors in their assessment that “future investigations are needed to understand the mechanisms underlying the opposite changes of synaptic connections in FrA and how such changes contribute to the acquisition, extinction and reinstatement of fear memories,” and add that future studies concurrently examining both RSC and FrA would be illuminating in better understanding their respective roles in fear conditioning.

6. When clustered new spines get eliminated after 4 – 6 weeks, do the whole cluster of spines disappear together? It will be informative if the authors can classify the pattern of elimination of clustered spines.

In analyzing spine stability at 4-6 weeks after learning we have shown that clustered spines are more stable than non-clustered spines added during learning (Fig 1k). The reviewer asks an interesting question regarding how clusters themselves, not just individual clustered spines, fare at this remote imaging point. We find that 32.3% of clusters are fully intact (i.e. all spines initially forming the cluster are still present at 4-6 weeks); 61.6% of clusters are partially intact with 1 or more of the clustered spines still present at the remote imaging point; and, only 6.1% of clusters are lost (i.e. all clustered spines are lost by 4-6 weeks post-training). These values were calculated as percentages for each animal as displayed in the graph with connecting lines indicating each animal. We add this result as Supplementary Fig. 9.

7. The authors claim that pre-learning spine turnover correlates with learning rate. Given the approximately sigmoidal shape of the learning curve shown in Suppl. Fig. 1a, it seems that the definition of the learning rate (slope) does not capture this feature, and it may be sensitive to fluctuations in freezing percentage after the animal's performance has almost plateaued. The authors may try to fit each mouse's learning curve with a sigmoidal curve and use the fitting parameters to correlate with learning performance.

We thank the reviewer for these suggestions. We analyzed our behavioral data using two methods for our correlations to spine dynamics. First, to avoid spurious fluctuations in freezing that are often seen as animals reach a behavioral plateau, we averaged freezing from all days of testing (day 2 to day 5, the first 45 seconds before shock is administered) for each animal. This is displayed in Fig1d and Fig1j. To explore other parameters of behavior, we also plot these correlations using a different method of analysis; here we calculated a learning rate for each animal which is essentially the slope of the line connecting each animal's baseline freezing with the peak of the each animal's plateau freezing, as described in our methods section. Finally, we display each animal's learning curve in our revised graph of contextual learning (Fig. 1c). In attempting to fit our behavioral data to a sigmoidal curve, two animals' freezing could not be properly fit, likely due to low overall freezing. As an alternative, we have also included a correlation of area under the curve of our freezing data, which similarly reflects overall learning. We add these new analyses to Supplementary Fig 1 in our revised manuscript.

8. The permutation test for spine clustering is problematic.

1) The first permutation test is invalid. The authors state that the “new spine” identity was randomly assigned to “the positions of all spines identified during imaging.” What is the rationale for this procedure? Why should the “new spine” identity be assigned to a stable spine or even an eliminated spine?

We appreciate the opportunity to clarify both the methods applied to and rationalization for utilizing resampling statistics in our clustering dataset. To begin, we found that training increased the percentage of new spines added during learning which occurred in clusters (i.e. new spines located within 5 μm of each other). This result was obtained by examining the percentage of new spine clustering during the learning phase in animals undergoing training compared with those that remained in their home cages. We were further interested in testing whether the percentage of clustering occurring with learning was significantly different from chance levels of clustering. To estimate chance clustering with the fewest *a priori* assumptions, we utilized permutation testing, wherein we resampled our existing dataset multiple times to find the distribution of percent new spine occurring in clusters if new spines were added at random. We used two different approaches to address this question and found similar distributions for random clustering. First, in each resampling, the number of new spines added per segment of dendrite was used to pick an equivalent number of random positions along the same segment and assess whether these positions were within 5 μm of each other (i.e. what percent of new spines when placed at random will occur within 5 μm and meet our clustering criteria?). This process was completed for all segments of dendrites for all animals for each resampling and repeated 10,000 times to provide the distribution displayed in Fig 1h for trained animals and Fig 1i for controls. To calculate an exact p-value we assessed how many resampled clustering percentages were as great or greater than our observed value and found none, which suggested to us that our observed rate of clustering associated with learning was significantly higher than the rate at which new spines would occur within 5 μm of each other if only governed by random addition of spines.

To confirm these findings we sought a similar but different approach for choosing positions along the dendrite to assess for clustering (i.e. positions within 5 μm of each other). In our first analysis we used the number of new spines added to a segment of dendrite to select an equivalent number of positions at random, anywhere along the same dendrite to assess for clustering. In our alternative approach we utilized the number of new spines added to a segment of dendrite to select an equivalent number of positions along the dendrite at random, but only utilizing positions where a spine had been observed (whether previously observed or still present). The subtle difference is that selected positions for each permutation were constrained to occur where spines were or had been in the dataset (Supplemental Fig 6a). Here, the rationale is that rules may exist that we are unaware of which govern where spines can possibly occur (i.e. there may be a location on a given segment of dendrite that is not able to have new spines added) and by selecting positions at which we know spines have occurred we would not violate these potential rules. To calculate an exact p-value we followed the same approach as above and assessed how many resampled percentages were as great or greater than our observed value and found none, which again suggested to us that our observed rate of clustering associated with learning was significantly higher than the rate at which new spines would occur within 5 μm of each other if only governed by random addition of spines. Also re-assuring was the finding that both methods of analysis yielded highly similar distributions of random clustering percentages.

2) It is unclear from the Methods section how the authors obtain the null distribution as shown in Fig. 1f, Fig. 3c, and Suppl. Fig. 6. According to the authors, the permutation test was done per dendritic segment. As spine density/distribution and the percentage of spine formation vary from segment to segment, the probability distribution of clustered spine formation will also differ from segment to segment, because with higher density of new spines there will likely be more clusters even under the null hypothesis. Thus it is invalid to pool the permutation results from all segments into a single histogram, and compare that with the overall percentage of clustered new spines.

In calculating clustering from our data, we utilized the total number of new spines occurring within 5 μm of each other divided by the total number of new spines stably added during learning to arrive at a percentage of clustered spines for each animal, and the mean of all animals was then calculated. We used a similar approach to calculate the percent clustering in our resampling analyses by proceeding from dendrite to animal to average of animals. Specifically, for a single animal each dendrite that contains new spines is resampled and the number of clustered spines is noted. When this is completed for all dendrites for a given animal the percent of clustered spines is calculated as above as the number of new spines within 5 μm divided by the total number of stably added new spines for that animal. In turn each animal's resampled clustering percentage is calculated and then these values are averaged together. This completes one resampling event, and this average value is then plotted on the distributions shown in figure 1,3 and supplemental figure 6. This process is then repeated over and over again for a total of 10,000 resamplings, which then gives the full distribution of random sampling. We would be pleased to openly share our R code utilized to calculate the resampling distributions with the reviewer and any member of the scientific community both as a resource and should someone wish to validate our calculations.

3) The second method of permutation test (Suppl. Fig. 6) seems reasonable. However, it yields a distribution of percentage of clustered new spines that appears inconsistent with the spine density and spine formation rate given in Suppl. Table 1 and Suppl. Fig. 3a. Take D0-2 control mice for example. The average spine density is ~ 0.4 spine/ μm , which means the average separation between spines is ~ 2.5 μm . The new spines gained from D0 to D2 account for $\sim 4\%$ of total spines on D0, which means that, if the new spines are distributed uniformly and independently along the dendritic segment as the authors assumed in the null hypothesis, their average distance would be ~ 2.5 $\mu\text{m}/4\% = 62.5$ μm . Under the null hypothesis, the new spines along the dendritic segment follow a Poisson distribution, and the inter-spine interval follows an exponential distribution with parameter $\lambda = 1/(62.5$ $\mu\text{m})$. Therefore, the probability that the nearest neighbor of a spine is within 5 μm (i.e., they form a cluster) is $1 - \exp(-5/62.5) \sim 7.7\%$, which is approximately the same as the number for trained mice given in Suppl. Fig. 4 and much higher than the number for control mice given there.

Please see below for our response about the calculation. Here, we wish to clarify that Supplementary Fig 4 in our previous manuscript is meant to demonstrate when, during training, newly added spines formed clusters. The reviewer is correct in noting that ~7% of spines are clustered by the second day of training. In our revised manuscript, we have expanded our explanation in the figure legend as well as including a diagram to illustrate how spine clustering was assessed (Supplementary Fig. 5).

Based on the same argument, if we take the total spine formation from D0 to D5 to be ~ 8% (Supple. Fig. 3) and ignore the complication caused by the fact that some spines formed during D0-2 were lost by D5, we will have the probability of clustering (again, irrespective of whether the spine was formed during D0-2 or D2-5) as $\sim 1 - \exp(-5/31.25) \sim 14.8\%$. This number, although an over-estimation, is much lower than that for control mice (29%, Suppl. Table 1).

Please see below for our response about the calculation. We realize that the reviewer used the total spine numbers in table 1 in the above calculations. We have also included the average clustered spines presented in the manuscript to this table for further clarity.

The authors need to give more details about the parameters and procedures used in the permutation test to clarify such discrepancies. If the control mice already exhibit significantly more clustering than expected from the null hypothesis, it would have a major impact on how the data for trained mice should be interpreted.

We again appreciate the opportunity to further clarify our methods. As mentioned, in an attempt to make the fewest *a priori* assumptions when assessing at what frequency spines would cluster at random, we utilized resampling analysis of our data. The reviewer astutely acknowledges that new spines should be added in a Poisson distribution under the null hypothesis (i.e. that spines are added at random) and utilizes average spine density and percent of spine addition to estimate the distance between new spines. We have completed these measurements for our data (i.e. the distance from each new spine to the nearest neighboring new spine) and find that trained mice have an average distance of 8.3 μm between new spines while control animals have an average distance of 13.6 μm (Fig. 1g, inset), considerably smaller for both than estimated by the reviewer. The reviewer identifies that if spines are added in a Poisson distribution, distance between spines (interspine distance) will follow an exponential distribution. In fact, we find that our cumulative probability distributions of interspine distances for trained and control animals closely match with values calculated from the exponential cumulative distribution function: $F(x|\mu) = 1 - e^{-x/\mu}$, where $\mu = 8.3$ for trained animals and $\mu = 13.6$ for controls (see cumulative probability distribution below). Further, given our data for trained animals we calculate, from the reviewer's suggestion, percent clustered spines $= 1 - e^{-(5/8.3)} = 0.452 = 45.2\%$, close to our observed average value of 42.0%. Similarly, for control animals, percent clustered spines $= 1 - e^{-(5/13.6)} = 0.307 = 30.7\%$, slightly larger than our observed value of 23.2%. A likely discrepancy between our measured values of interspine distance and average distance between new spines and that estimated by the reviewer is that we observed that not all segments of dendrite have new spines added. Thus, there is not a perfectly uniform distribution across all segments of dendrite as was assumed in the reviewer's estimation. Further, our resampling analysis is not hampered by this fact that spines are not added to all segments of dendrite, as our analysis occurred only for those segments which did in fact have new spines added. In sum, our measured interspine distances between new spines are consistent with expectations given a Poisson distribution, though as the reviewer points out, our calculated distributions are quite different from those estimated by uniform spine addition along all lengths of dendrite. We believe this further supports our hypothesis that new spines are not added in a uniform fashion during learning.

9. The idea that hotspots of spine turnover have higher clustered spine formation is very interesting. However, there are a number of issues with the analysis.

1) In this analysis, do spines that are eliminated count towards the “spines having undergone turnover”?

We define turnover as spines that are gained and lost between imaging sessions, and thus count eliminated spines in our measurements of turnover. If a spine was not eliminated it would be considered stable in our analysis and not factor into spines having undergone turnover.

2) In Fig. 3b there are apparently 3 categories: segments with pre-learning turnover but without learning-related clustered spines (data points on x-axis), segments with learning-related clustered spines but without pre-learning turnover (data points on y-axis), and segments with both. How many segments are there in each category? Does the distribution among these categories differ from the null hypothesis, i.e., clustering is independent of pre-learning turnover?

We appreciate the reviewer's suggestion to analyze further the distribution of various types of dendritic segments. In fact, Fig 3b shows 4 categories of segments, the 3 types mentioned by the reviewer, as well as segments lacking both turnover and clustering, which fall at the origin. We have added the analyses of distribution of segments in these 4 categories, and null hypothesis simulations in our revised manuscript (see Supplementary Fig. 16). The null hypothesis simulation is done by permuting the number of clustered spines on each dendritic segment within each animal and recalculating the percentages of segments of the 4 categories. For example, a mouse has 10 segments and each segment has 2 numbers: number of clustered spines, and number of turnover spines. We simulate the null hypothesis by shuffling the number of clustered spines on the 10 segments, without changing the original number of turnover spines on each segment. This will yield a random distribution of clustered spines on dendritic segments that is independent of pre-learning turnover. The result shows that the observed distributions among 4 categories are significantly different from the simulated values. Therefore, this finding further supports our model that clustering and turnover are related process.

3) How did the authors generate the null model for Fig. 3c? What does "random chance" mean in this simulation?

We appreciate the opportunity to further clarify our resampling/randomization analyses. Similar to our prior resampling analyses, we utilized two methods to determine "random chance." In each case, we attempted to determine the average distance from each clustered spine to the nearest pre-learning turnover spine. To do this, we randomized the positions of pre-learning turnover spines, assessed the average distance to the nearest clustered spine and then repeated this process 10,000 times. In one method (now shown in Supplementary Fig. 6b), we made use of the locations of all observed spines as potential sites for randomization of pre-learning turnover spines (excluding the known locations of clustered spines). In the alternative method (now shown in Fig. 3c), pre-learning spines were randomly assigned to any location along the dendrite during each resampling. Again, both methods yield remarkably similar distributions. We have included these additional details concerning this resampling analysis in our revised methods.

4) In control mice there are also clustered spine formation. Do these clusters also tend to form near sites with previous turnover? What is the average distance to nearest turnover site in control mice? Is it comparable or significantly different from the distance in learning mice?

We have conducted this analysis and find that the average distance from a clustered spine to the nearest turnover site is significantly larger in control animals compared to trained animals (Supplementary Fig. 15c).

5) Are sites of clustered spines themselves more dynamic? If so, this would imply that hotspots tend to persist.

We thank the reviewer for the interesting question. Our model of spine clustering and spine turnover hotspots posits that regions of turnover (i.e. spine gain and loss) are the locations in which clustering is mostly likely to happen as a means to stabilize new synaptic contacts. Thus, regions of spine clustering are intrinsically dynamic given the concomitant turnover that is occurring. The question of the localization of hotspots and clustering over time is a truly interesting one. We would suggest that knowing where these phenomena are occurring within the dendritic tree would give us insight into specific areas of in which new memories are being stored. A related and intriguing question is how this localization of turnover and clustering dynamically changes with time. Fu *et al.*'s 2012 study on motor learning induced clustering suggests that as animals learn different tasks clustered spines related to each task are independent (i.e. themselves not spatially clustered). This suggests to us that localization of turnover might also be independent across learning tasks and would mean that hotspots are not static locations along the dendrite, but are dynamically changing. We have not attempted to cross train animals on MWM and CFC as stress associated with each task can have interfering effects on the other (Conrad 2010); however, utilizing a conceptually similar procedure to Fu *et al.* would be one method to attempt to answer this question in future studies.

10. The authors need to provide much more details about the biophysical model (Fig. 4) for the reader to understand how it works and how it is related to other results presented in this manuscript.

We thank the reviewer for this suggestion. We have now included a detailed description of the computational model in the Supplementary Experimental Methods. We have also made a number of changes from the initially submitted model, which we believe make the model more relevant to the experimental results presented in the manuscript.

Minor issues:

1. The term “gained spines” is confusing, as it could mean spine formation, or the net number of spines gained (i.e., formation minus elimination).

We initially utilized the term gained spines to mean the pool of spines that were newly formed for any given observation period, not the net number of spines gained (i.e., formation minus elimination). We are sorry that this terminology was confusing and have made the appropriate changes. All instances of the term “gained spines” have been changed to “new” or “newly formed” spines in our revised manuscript.

2. The actual values of spine dynamics are often shown as plots but not explicitly given in numbers.

We have added numerical values to spine dynamics where applicable in figure legends.

3. Is there any mouse that was trained but failed to learning the task? If so, did they exhibit similar spine dynamics/clustering as control mice?

We did not observe any mice that failed to learn the task. There are examples of animals that have lower levels of average freezing and these animals also have lower levels of clustering, a result consistent with our finding that there is a strong correlation between spine dynamics/clustering and performance in contextual and spatial learning tasks.

Reviewer #2 (Remarks to the Author):

Frank and colleagues follow dendritic spine dynamics in retrosplenial cortex during contextual fear conditioning and Morris Water Maze learning. They find that both the amount of pre-learning spine turnover as well as clustering of new spines during learning are correlated with learning performance. They further find that a transgenic mouse model that has previously been demonstrated to show enhanced learning also has higher pre-learning spine turnover and more learning-related clustered new spines.

This is an interesting and timely study, the data seem solid and mostly carefully analyzed. My strong concern is the model, which is very opaque. It is difficult to judge its quality and validity, as far too little information is provided: how exactly was the model adapted for the study? What had to be changed to fit the data? The presentation of the model results seems misleading. The methods say: “Clustering was defined as the percentage of dendritic branches containing 3 or more potentiated synapses.” This would mean that there is simply more synaptic input, not more clustered input, which is trivial as spine turnover was increased? The authors should plot percentage of clustered new synapses of total gained synapses as in Figure 1d and g, or if the model does not contain addition of new spines, percentage of clustered potentiated synapses of all potentiated synapses within branches. But if the model only contains potentiation of synapses rather than addition of new synapses, and/or if there is no modelling of the spatial arrangement of synapses within branches, how does the model relate to the presented data? In my opinion it would be better to remove the model in its present state, the experimental data are strong enough on their own.

We thank the reviewer for these constructive comments and suggestions. We have made a number of changes to the initially submitted model and its description in the manuscript, which we believe more strongly relate the computational model to the experimental data presented in the paper. Specifically:

- 1) We have now extended the initially submitted model to include the spatial arrangement of synapses throughout the length of dendritic subunits. Our updated model contains a newly introduced rule for distance-dependent synaptic capture.
- 2) The spatial arrangement of potentiated synapses after learning is now used to assess the degree of clustering after learning instead of the count of synapses per branch. We used a criterion for clustering that is similar to the one used in the experiments: Pairs of synapses that were less than 10 μ m from each other are considered clustered.
- 3) We have replaced the plot in Fig. 4a with a new plot that shows clustered pairs of synapses as a percentage of the total pairs of synapses in dendritic subunits. This percentage increases as the degree of synaptic turnover increases in the new Fig 4a.
- 4) We have plotted the percentage of clustered *new* synapses, as suggested by the reviewer, in Fig. 4b. We considered the ones which were added and potentiated beyond the first day of simulated learning as “new” synapses.
- 5) We have expanded the supplementary experimental methods with a detailed description of the computational models that we used and the addition of Supplementary Table 3 which lists all the parameters of the model.

We also point out that our model already simulated synaptic turnover of older weak synapses. As such, we believe the updated model reinforces the experimental findings of the paper: we find that increasing synaptic turnover increases the clustering of synapses in branches in general, and that new synapses tend to be clustered, in line with the experimental evidence. We additionally show that the increase of turnover leads to an increase in the sparsity of memory engrams, which could be related to greater memory capacity. These points have been added to the main text and the legend of Figure 4 which has now been updated. The computational model is described in detail in Supplementary Experimental Methods.

Further points:

- The analysis in Figure 3b is not suited. Because turnover and clustering are related at the level of individual mice, and mice have different levels of turnover/clustering, plotting the values of individual dendritic segments of all mice together without z-scoring will automatically result in a positive correlation. If the authors want to make the point that that individual dendrites with more turnover also have more clustering, it is imperative to show this WITHIN mice, for instance by z-scoring to normalize data between mice.

We appreciate the reviewer's careful reading of our manuscript and request that we normalize pre-learning turnover and spine clustering in our analysis of the correlation between these data. We would like to clarify that we have plotted both turnover and clustering per 100µm of dendrite in Fig. 3b, not directly counts or percentages of turnover and clustering. We utilized this method to normalize across all dendritic segments. To confirm these findings we have further included two alternative methods of calculating clustering and turnover at the segmental level. First, we calculated the percentage of clustering/turnover present on a given segment as a percentage of the total clustering/turnover in a given animal. In this way, all segments added together for a particular animal for clustering/turnover equal 100%. These data are shown in supplementary figure 15a & d. Further, we also calculated the percent of clustering/turnover on each segment (i.e. the number of clustered spines/total number of new spines on each segment; the number of turnover spines/the total number of spines over the days which turnover was observed) and standardized these values utilizing z-scoring ($z\text{-score} = (\text{segment value} - \text{animal mean}) / \text{animal standard deviation}$). These data are shown in supplementary figure 15b & e.

- Do animals with more pre-learning spine turnover also have increased spine gain/loss during learning? Is learning performance therefore related to spine gain, loss or turnover during learning? This is important to be able to make the point that it is really the PRE-LEARNING turnover that is important for learning.

The reviewer asks an interesting and insightful question about our data; namely, is pre-learning spine turnover unique in its relationship to learning or are other aspects of spine dynamics during learning also features that influence learning. We have shown that pre-learning spine turnover (i.e. spines gained and lost before learning) is significantly positively correlated with future learning (Fig. 1d). We also show that spine turnover itself is not influenced by learning (Supplementary Fig. 4c) and now include data showing that pre-learning spine turnover is significantly correlated to spine turnover during learning (Supplementary Fig. 8c). Thus, animals with higher turnover before learning also have higher levels of spine turnover during learning, and this turnover is also correlated with average freezing (Supplementary Fig. 8d). While prior data had shown changes in spine dynamics during learning, particularly in relationship to pre-learning spine dynamics (Roberts *et al.* 2010), we do not find this result in our data, which we believe is related to both our region of interest (RSC) and the fact that we are *not* examining spine dynamics during critical periods of plasticity in early development, as in Roberts *et al.* During learning we do not find a significant correlation between average freezing and total spine gain (Supplementary. Fig 8a). However, we do find a significant correlation between spine loss during learning and average freezing (Supplementary. Fig 8b), which we feel is a fascinating result. We believe this is due to the relationship between spine loss and clustered spine formation we have found; specifically, we find that the nearest neighbor distance between lost spines and their closest gained neighbor is significantly smaller if the neighboring gained spine is a clustered spine (Fig. 3g). This suggests to us that spines are lost near clustered gained spines and supports our hypothesis of hotspots of spine turnover.

- Is the turnover and clustering only related to the speed of learning or also to how well mice do at the end of training? Please plot the data in Figure 1c and g separately for freezing at day 2 and day 5.

We have plotted these data as requested by the reviewer (Supplementary Fig. 3). However, we do not find a significant correlation between clustering and freezing at day two. Further, we do not find a significant correlation between freezing at day 2 or day 5 and pre-learning turnover. There is a significant correlation between clustering and day 5 freezing. We chose to plot average freezing over the course of training against either clustering or pre-learning turnover to overcome fluctuations in freezing that are typically seen when animals reach asymptotic levels of freezing.

- Related to the above question: please provide a time course of performance over learning for the Morris water maze for WT and transgenic mice.

We added this graph as Supplementary Fig. 11c in our revised manuscript. CCR5 mutant mice learned faster than WT littermates.

- Clustering analysis: how was the distance between spines determined? Is it the distance between spine bases along the

dendrite? Please add to methods.

In assessing clustering, we measured the distance from the middle of one spine base to the next. The measurement followed the shape of the dendritic shaft. This was added to our revised methods.

Reviewer #3 (Remarks to the Author):

In the present manuscript, Frank, et al. reported learning-related clustering of spine in the retrosplenial(RSC) and its relationship with pre-learning turnover of spines using thy1-YFP expressing H-line mice. They found that the fear conditioning was more effective in those animal with higher pre-learning spine turnover, and induced clustered spine generation, which occurred more frequently in the better learning mice. The authors test the relationship using Ccr5 hetero KO mice (Ccr5) which showed rapid fear conditioning. They found both pre-learning spine turnover and post-learning-clustering were enhanced in Ccr5. Clustered new spines were more persistent than non-clustered one in WT and Ccr5. The authors confirmed the result with the Morris water maze. The clustered spine generation was more frequently detected in those mice with larger pre-learning spine turnover in WT mice. These results could be explained by their model in a previous study (Kastellakis et al., Cell Rep. 2016). The model also predicted, more sparse coding and less overlap of neurons in a neuronal network. The authors conclude such hotspot serves to modulate network function, and influencing storage capacity and learning and memory. These observations are intriguing and valuable to the field, provided that the major problem below is experimentally resolved.

The authors overlooked the fact that spine turnovers were increased in many animal models of ASD, where learning is impaired (ex. Pan et al. PNAS 107:17768, 2010; Padmashri et al. J. Neurosci. 33:19715, 2013; Isshiki et al. Nat Commun 5:4742, 2014; Jiang et al. J. Neurosci. 33:19518, 2013). These are opposite to the finding in this manuscripts. Thus, the authors need to reconcile their observations with the existing knowledge. One key observation is that spine turnovers are caused not only by plasticity, which is activity-dependent, but also by intrinsic fluctuations of spine structures, which are independent of any neuronal firing and plasticity mechanisms (Yasumatsu et al, J. Neurosci. 28:13592, 2008; Minerbi et al. PLoS Biol. 7:e1000136, 2009; Loewenstein et al. J. Neurosci. 35:12535, 2015; Dvorkin et al. PLoS Biol. 14:e1002572, 2016). Increased spine turnovers seen in ASD models were ascribed not to plasticity but to fluctuations in the case with FMRP1-KO mice (Nagaoka et a., Sci. Rep. 6:26651, 2016). This work showed that plasticity-induced spine turnovers were effectively prevented by intraperitoneal infusion of MK801 twice daily (0.25 mg/kg).

To resolve the major discrepancy, the authors should examine whether the MK801 treatment restores the enhanced spine turnovers in Ccr5 mice during the pre-training period, and those in WT during learning tasks as a positive control. If the increase in the turnovers are restored by MK801, it shows increase in baseline learning, and is consistent with the authors interpretations. Such results also provide a way to understand why spine turnovers induce both the impairment and enhancement of learning in ASD and Ccr5 mice, respectively. The authors should provide a full introduction and explanation to the apparent discrepancy between ASD model and Ccr5 mice. If the increased turnovers might not be blocked by the MK801 treatment, the authors need to find other ways to consistently interpret their data sets and existing references.

We thank the reviewer for bringing up this important issue. We followed the reviewer's suggestion and performed MK801 injections to CCR5 mutant and WT mice before and throughout learning days. Spine images were taken with the same schedule as previous CFC experiments in Fig 1 and 2. MK801 indeed prevented the enhanced spine turnover of Ccr5 mutant mice, while not having a significant effect on WT turnover. This suggests that the enhanced spine turnover caused by the Ccr5 mutation is due to plasticity-dependent mechanisms, therefore differing from Fmr1 knockout mice. MK801 also impaired spine clustering after learning in both CCR5 mutant and WT mice, which is consistent with our model that spine clustering is learning- and plasticity-dependent, and is likely to be a memory storage mechanism. We added this result as Supplementary Fig. 10, and as a paragraph in the discussion of our revised manuscript.

Minor points:

1) The size of clustering is only qualitatively described in line 76 (two or more spines within 5um of each other) and rest of the manuscript. Could the authors be more quantitative about the size of cluster, for comparison in the future experiments, as the cluster is the major concern in this manuscript?

We agree with the reviewer that clusters of spines are a primary focus of our manuscript. As such, we have included a thorough summary of clustering properties in Supplementary Table 2.

2) In figure 3, the authors examined only WT data, but the same analysis should be performed for Ccr5 mice.

We have performed the parallel analyses as the reviewer suggested, and added in Fig. 3 in our revised manuscript. In Ccr5 mutant mice, pre-learning turnover also predicts learning-related clustering, at both mouse and dendritic segment level. The closest distance between clustered spines to turnover spines is also significantly smaller than the simulated value under the null hypothesis. Therefore, our Ccr5 results are consistent with the WT data.

3) In figure 4, the authors try to explain the experimental data of real dendritic spines with a theoretical model where only very simple mechanisms are incorporated into small circuits. Just for example, the model ignores the existence of synapse

fluctuations. In other words, only way to account for the increase in spine turnover in this model is plasticity. Another point is that Fig. b-d must be pure predictions. In order to distinguish experimental data from the simplistic model predictions, the authors may consider moving Fig. 4 to supplementary figure, particularly because the model does not incorporate the fluctuations, although I think the model simulation is still valuable.

We thank the reviewer for this comment. We first would like to point out that synaptic plasticity is not the only mechanism for reorganization in our model circuit, but we have purposefully included a mechanism for synaptic turnover, which acts on weak synapses, replacing them with newly formed weak synapses from random presynaptic inputs. This is based on the experimental evidence which suggests that large (potentiated) spines are persistent and more stable than small synapses or filopodia [1][2]. On the other hand, we agree with the reviewer that synaptic plasticity is the mechanism that guides synaptic clustering. Although we don't explicitly model synapse fluctuations in all synapses, the turnover mechanism allows us to simulate to a degree fluctuations for weak synapses.

We have updated Fig. 4 with results from the updated version of our model that incorporates explicitly the cooperative plasticity of neighboring spines and we have updated all the panels of the figure. We believe the new modeling results presented in figure 4a and 4b are directly relevant to the experimental results of the paper. The description of the model in the Supplementary Experimental Procedures now contains a detailed description of the methods and models that we used to simulate synaptic plasticity and synaptic turnover.

[1] J. T. Trachtenberg, B. E. Chen, G. W. Knott, G. Feng, J. R. Sanes, E. Welker, and K. Svoboda, "Long-term in vivo imaging of experience-dependent synaptic plasticity in adult cortex.," *Nature*, vol. 420, no. 6917, pp. 788–94, 2002.

[2] A. J. G. D. Holtmaat, J. T. Trachtenberg, L. Wilbrecht, G. M. Shepherd, X. Zhang, G. W. Knott, and K. Svoboda, "Transient and persistent dendritic spines in the neocortex in vivo.," *Neuron*, vol. 45, no. 2, pp. 279–91, Jan. 2005.

4) In line 32 and 160, the authors cite a paper (*Annu Rev Physiol* 55:397, 1993) for spine plasticity, but it gives a kind of historical consideration on structural plasticity, mainly of presynaptic terminals, and may be out of place. There is a more appropriate review, where notions of plasticity and intrinsic fluctuations of dendritic spines are consistently described (*Trends Neurosci.* 33:121, 2010).

We thank the reviewer for this updated and highly appropriate reference; we have included it in our revised manuscript.

REVIEWERS' COMMENTS:

Reviewer #1 (Remarks to the Author):

The authors have addressed most of our questions satisfactorily and significantly improved the clarity of the manuscript.

Only one major concern remains. I still believe that it is important to use either context or foot shock per se as a control condition for the CFC experiment. I understand that either of these conditions alone may suffice to increase spine formation, but they do not necessarily increase clustered spine formation. Therefore, they could provide be the ideal control condition to demonstrate that the observed clustered spine formation is learning specific.

Also, two minor points:

1. The authors provide nice explanation on why RSC differs from PFC in spine dynamics in response to CFC (Answer to Reviewer 1's Q5). It will be nice to include this in the Discussion section of the manuscript to highlight the heterogeneity in different brain regions and in imaging paradigms, which necessitates caution when generalizing and comparing results from different brain regions.

2. The authors added Suppl. Fig. 9 for the fate of spine clusters. I have a simple additional question: does the observed fate of clusters match the expectation based on the fate of individual clustered spines? Intuitively, larger clusters are more likely to persist at least partially. If the probability of elimination for each clustered spine is p , and if each clustered spine's fate is independent of other spines in the cluster, the probability of the cluster to be completely lost is p^n , where n is the number of spines in the cluster. The probability of the cluster to persist partially with m out of the n initial spines remaining is $C(n \text{ choose } m) * p^{(n-m)} * (1-p)^m$, and so on. Such analysis will reveal whether the size of the cluster affects the survival of constituent spines.

Reviewer #2 (Remarks to the Author):

The authors addressed my concerns and I am fine with the revised manuscript.

Reviewer #3 (Remarks to the Author):

The authors have very nicely addressed all my previous concerns.

Reviewer #1 (Remarks to the Author):

The authors have addressed most of our questions satisfactorily and significantly improved the clarity of the manuscript.

Only one major concern remains. I still believe that it is important to use either context or foot shock per se as a control condition for the CFC experiment. I understand that either of these conditions alone may suffice to increase spine formation, but they do not necessarily increase clustered spine formation. Therefore, they could provide be the ideal control condition to demonstrate that the observed clustered spine formation is learning specific.

(Not required by editorial team)

Also, two minor points:

1. The authors provide nice explanation on why RSC differs from PFC in spine dynamics in response to CFC (Answer to Reviewer 1's Q5). It will be nice to include this in the Discussion section of the manuscript to highlight the heterogeneity in different brain regions and in imaging paradigms, which necessitates caution when generalizing and comparing results from different brain regions.

We thank the reviewer for this suggestion. In our revised manuscript, we have included the discussion about the heterogeneity of results from different brain regions and behavioral paradigms.

2. The authors added Suppl. Fig. 9 for the fate of spine clusters. I have a simple additional question: does the observed fate of clusters match the expectation based on the fate of individual clustered spines? Intuitively, larger clusters are more likely to persist at least partially. If the probability of elimination for each clustered spine is p , and if each clustered spine's fate is independent of other spines in the cluster, the probability of the cluster to be completely lost is p^n , where n is the number of spines in the cluster. The probability of the cluster to persist partially with m out of the n initial spines remaining is $C(n \text{ choose } m) * p^{n-m} * (1-p)^m$, and so on. Such analysis will reveal whether the size of the cluster affects the survival of constituent spines.

We thank the reviewer for this suggestion. In our revised manuscript, we have now included supplementary table 3 that contains this information. Using the 2-spine cluster as an example, the probability of the cluster to be completely lost is $(32/78)^2=16.8\%$, which is close to the observed value of lost clusters (6/39=15.4%). The probability of the cluster to be partially stable with 1 spine remaining is $C(2 \text{ choose } 1) * (32/78)^{(2-1)} * (1-32/78)^1=48.4\%$, which is close to the observed value of partially stable clusters (20/39=51.3%). We utilized Fisher's exact test to test the independence of spine or cluster stability and cluster size and found that both rates of cluster loss and spine survival are independent of cluster size.

Reviewer #2 (Remarks to the Author):

The authors addressed my concerns and I am fine with the revised manuscript.

Reviewer #3 (Remarks to the Author):

The authors have very nicely addressed all my previous concerns.